# Heliorhodopsin-mediated light-modulation of ABC transporter

Shin-Gyu Cho [1,2,3], Ji-Hyun Kim [1,3], Ji-eun Lee[1], In-Jung Choi[1], Myungchul Song[1], Kimleng Chuon [1], Jin-gon Shim[1], Kun-Wook Kang[1] & Kwang-Hwan Jung [1] ✉

Heliorhodopsins (HeRs) have been hypothesized to have widespread functions. Recently, the functions for few HeRs have been revealed; however, the hypothetical functions remain largely unknown. Herein, we investigate light-modulation of heterodimeric multidrug resistance ATP-binding cassette transporters (OmrDE) mediated by *Omithinimicrobium cerasi* HeR. In this study, we classifiy genes flanking the HeR-encoding genes and identify highly conservative residues for protein–protein interactions. Our results reveal that the interaction between OcHeR and OmrDE shows positive cooperatively sequential binding through thermodynamic parameters. Moreover, light-induced OcHeR upregulates OmrDE drug transportation. Hence, the binding may be crucial to drug resistance in *O. cerasi* as it survives in a drug-containing habitat. Overall, we unveil a function of HeR as regulatory rhodopsin for multidrug resistance. Our findings suggest potential applications in optogenetic technology.

Rhodopsins are a light-driven seven-transmembrane protein with a retinal chromophore that binds to a specific Lys residue, and are found in prokaryotes, eukaryotes, and viruses[1,2]. Rhodopsins are classified into two types; type-II animal rhodopsins function as a G protein-coupled receptor in visual and non-visual phototransduction, whereas type-II microbial rhodopsins serve as ion pumps, ion channels, and light sensors[3]. In the marine and terrestrial microbial metagenomes, 48% of microorganisms harbored the microbial rhodopsin genes[4]. A recent report suggested that microbial rhodopsins are a dominant photoheterotrophic pigment to harvest solar energy in the ocean[5].

Heliorhodopsins (HeRs) are a rhodopsin discovered 5 years ago, forming a distinct clade separate from type-I rhodopsins. The topology of HeRs is inverted, as compared to that of type-I and -II rhodopsins, with the N- and C-termini located on the cytoplasmic and extracellular sides, respectively[6,7]. Bacterial HeRs lack ion pumping activity and have a long-lived photoactivated state, as do sensory rhodopsins (SRs) that belong to type-I[6-8]. In the same operon encoding SR, a single promoter can transcribe transducer protein and SR-encoding genes, and SR interacts with the transducer protein[9]. Researchers have hypothesized various functions of HeRs, such as influencing signal transduction in

the organism, interacting with adjacent genes in an operon, and enzyme activity via an enzyme domain fused to HeRs[6-8,10,11]. Although numerous hypotheses have been proposed, they have not yet been tested experimentally.

Recently, Cho et al. and Shim et al. described the functional roles of HeRs in modulating with enzymes adjacent to *helR* (HeR-encoding gene) in the same operon[12,13]. In addition, Hososhima et al. described that viral HeR proton-transport activity could be observed using a whole-cell patch-clamp[14]. Despite these findings, the functions of HeRs remain largely unknown. Thus, we hypothesized that other proteins might also interact with HeRs and analyzed microorganisms harboring *helR*. In this study, *helR* and genes adjacent to *helR* in the same operon were predicted to be continuously transcribed by a single promoter. After analysis, HeRs were classified into ten groups based on their adjacent genes. Among these ten groups, we focused on HeRs belonging to the group of ATP-binding cassette transporter (ABCT)-containing operons.

ABCTs are a membrane protein belonging to a large superfamily found in all kingdoms of life; they transport compounds such as nutrients, lipids, sterols, and drugs across the phospholipid bilayer.

[1]Department of Life Science, Sogang University, Seoul, South Korea. [2]Research Institute for Basic Science, Sogang University, Seoul, South Korea. [3]These authors contributed equally: Shin-Gyu Cho, Ji-Hyun Kim. ✉e-mail: kjung@sogang.ac.kr

ABCTs comprise four domains: two transmembrane domains (TMDs), which form a translocation pathway, and two nucleotide-binding domains (NBDs), which hydrolyze ATP and drive conformational changes in the TMDs to facilitate transmembrane transport[15]. Depending on whether the compound is being transported inward or outward, ABCTs are classified as an importer or exporter. ATP-binding cassette (ABC) importers, which are unique to bacteria, are involved in nutrient uptake. Their TMD and NBD subunits form individual chains. On the contrary, ABC exporters, which exist in both eukaryotes and prokaryotes, play diverse roles in transporting lipids, sterols, and drugs. Their TMB and NBD subunits are fused[15]. Protein sequence similarities and oligomeric forms in ABCTs were considered to classify ABCT[16–22].

In this work, we investigate HeR-mediated light-modulation in the relationship between *Omithinimicrobium cerasi* strain USBA17B2 HeR (OcHeR) and heterodimeric multidrug resistance ABCTs (OmrDE). Interaction positions between OcHeR and OmrDE are investigated using protein–protein docking simulation and isothermal titration calorimetry (ITC) analysis. Afterward, we determine how the interaction affects OmrDE activity in vitro and in vivo in the absence and presence of light.

## Results

### HeR-encoding genes analysis

We analyzed the operons containing *helR* and the neighboring genes in various organisms to investigate their relationship. *helR* was identified in many organisms and was noted to be distributed across Actinobacteria, Archaea, Chloroflexota, Baciliota, viruses, and unsorted group classes. Among these organisms, 38.6% of both *helR* and the neighboring genes in the same operon were predicted to be transcribed by a single promoter (Fig. 1a). This pattern was not predominantly identified in any specific class but in several classes. Particularly, the pattern within the phylum Actinobacteria class was 18.8%; however, no virus pattern was identified, suggesting that these patterns occur only in prokaryotes (Fig. 1a). Frequently, the neighboring genes flanking *helR* were identified in the same operons. Thus, we classified HeRs into ten groups with each of frequent neighboring genes: glutamine synthetase, NAD⁺ synthetase, photolyase, MerR family transcriptional regulator, TerC family protein, ABCT, SDR family NAD(P)-dependent oxidoreductase, alpha/beta hydrolase, domain unknown function (DUF) 2177, and DUF2238 (Fig. 1b, Supplementary Data 1–10), suggesting that the HeRs in each group of the frequent neighboring gene-containing operons potentially modulate proteins.

Moreover, we identified the residues responsible for interaction with proteins encoded by the neighboring genes. We analyzed the differences of amino acid frequency per HeR at each position in OcHeR by studying the amino acids of each HeR in a co-transcription group (*helR* and the neighboring gene in operon) and the classified ten groups to a non-co-transcription group (*helR* alone in operon) (detailed analyzing procedures are in Supplementary Fig. 1). In the co-transcription group, when compared with the non-co-transcription group, specific residues were not observed (Supplementary Fig. 2), while in classified ten groups, when compared with the non-co-transcription group, significant differences were observed (Fig. 1c, Supplementary Fig. 3). Particularly, the number of amino acids in hydroxylic, aliphatic, aromatic, acidic, and basic groups increased. Notably, aromatic residues participate in protein structure and protein hydrophobic interaction, whereas acidic and basic residues are crucial in polar interaction. Therefore, we suggest that the classified ten groups with increased specific residues are more likely to exhibit protein–protein interactions (PPIs) than the non-co-transcription group.

PPIs involve the proximity of two proteins via translational and rotational diffusions. An association rate constant enhanced by a long-range force (5–10 Å, electrostatic interaction) is stronger than that enhanced by short-range forces (hydrophobic and Van der Waals), accelerating diffusion due to electrostatic attraction between proteins[23]. Charges can affect an electrostatic interaction and be specific to each protein[24]. The "positive-inside" rule for membrane proteins is that positively charged residues are more frequently found on the cytoplasmic side of membrane proteins than on the extracellular side[7,25]. The charged residues of HeRs are important for binding to enzymes[12,13]. Considering the characteristic of electrostatic interaction, we suggest that charged residues in OcHeR may affect PPIs. The positive charges in OcHeR were distributed on the cytoplasmic side, consistent with the "positive-inside" rule (Fig. 1d). Twenty-two charged and aromatic residues were located on the cytoplasmic side in OcHeR homodimer (Supplementary Fig. 4).

Furthermore, we analyzed positively and negatively charged residues on the cytoplasmic side in HeRs from data in Fig. 1a, revealing dominant positively charged residues in ICL1 and ICL3. Particularly, the charged residues in R229, R233, R235, and D236 positions of OcHeR exhibited highly conservative charged residues (Fig. 1e). When comparing the distribution of charged residues between the non-co-transcription and the co-transcription groups, no significant differences were observed (Supplementary Fig. 5a, b). In the co-transcription group, all classes showed highly conserved positively charged residues (Supplementary Fig. 5c–g). Overall, HeRs showed a greater conservation of positively charged residues than that of negatively charged residues, suggesting the importance of positive residues in facilitating PPI.

### Characterization of OcHeR and OmrDE

We focused on a group of ABCT-containing operons because HeRs may interact with ABCTs as membrane proteins similar to SR interacting with membrane proteins. OcHeR, one of the HeRs in the group of ABCT-containing operons, was determined to belong to a distinct HeR clade through the phylogenetic tree and alignment of rhodopsins (Fig. 2a, Supplementary Fig. 6a, b). In the group of ABCT-containing operons, adjacent genes were encoded upstream of *helR*, suggesting that the proteins of these adjacent genes interact with HeRs. A promoter in the operon of the *O. cerasi* strains USBA17B2 was predicted to be upstream of *hp*, which may transcribe four genes (Fig. 2b).

To determine the type of ABCTs involved, we analyzed the protein structures and sequences of *omrD* and *omrE*. These genes encode two half-site ABCTs that are clustered and are predicted to be regulated by a single promoter, suggesting that they form heterodimeric ABC complexes (OmrDE; Fig. 2b). A protein structure of OmrDE based on TmrAB as a template was predicted using Swiss-Model (details are provided in the Supplementary Discussion)[15,26–28]. We compared the protein structures of OmrDE and ABC importers and exporters, observing that OmrDE was similar to type-I ABC exporters, which mediate the transport of highly hydrophobic substrates such as drugs and phospholipids (Supplementary Fig. 7a, b). In terms of structural basis, the protein structures between OmrDE and TmrAB had similar structures of TMDs and NBDs, except for the extracellular loop in TMD of OmrE (Fig. 3a). Furthermore, functionally important motifs—which participate in the catalytic cycle of ATP binding/hydrolysis and driving conformational changes of TMDs—were structurally similar between OmrDE and TmrAB, also NBD-defined motifs including Walker A, Walker B, ABC signature, A-loop, Q-loop, D-loop, and H-loop were highly conserved in OmrD and OmrE (Fig. 3b, Supplementary Fig. 6c and 7c)[29]. Particularly, the canonical catalytic glutamate next to the Walker B motif in OmrE, the ATP-binding site in the NBD, is conserved. The degrees of protein similarity and identity between OmrD as well as OmrE and the multidrug ABCTs were 62–73% and 25–40%, respectively (Supplementary Fig. 8a). In contrast, catalytic glutamate was replaced by aspartate in OmrD, leading to a non-canonical ATP-binding site (Supplementary Fig. 6c). In the phylogenetic tree of multidrug ABC exporters, OmrD and OmrE formed heterodimer clades

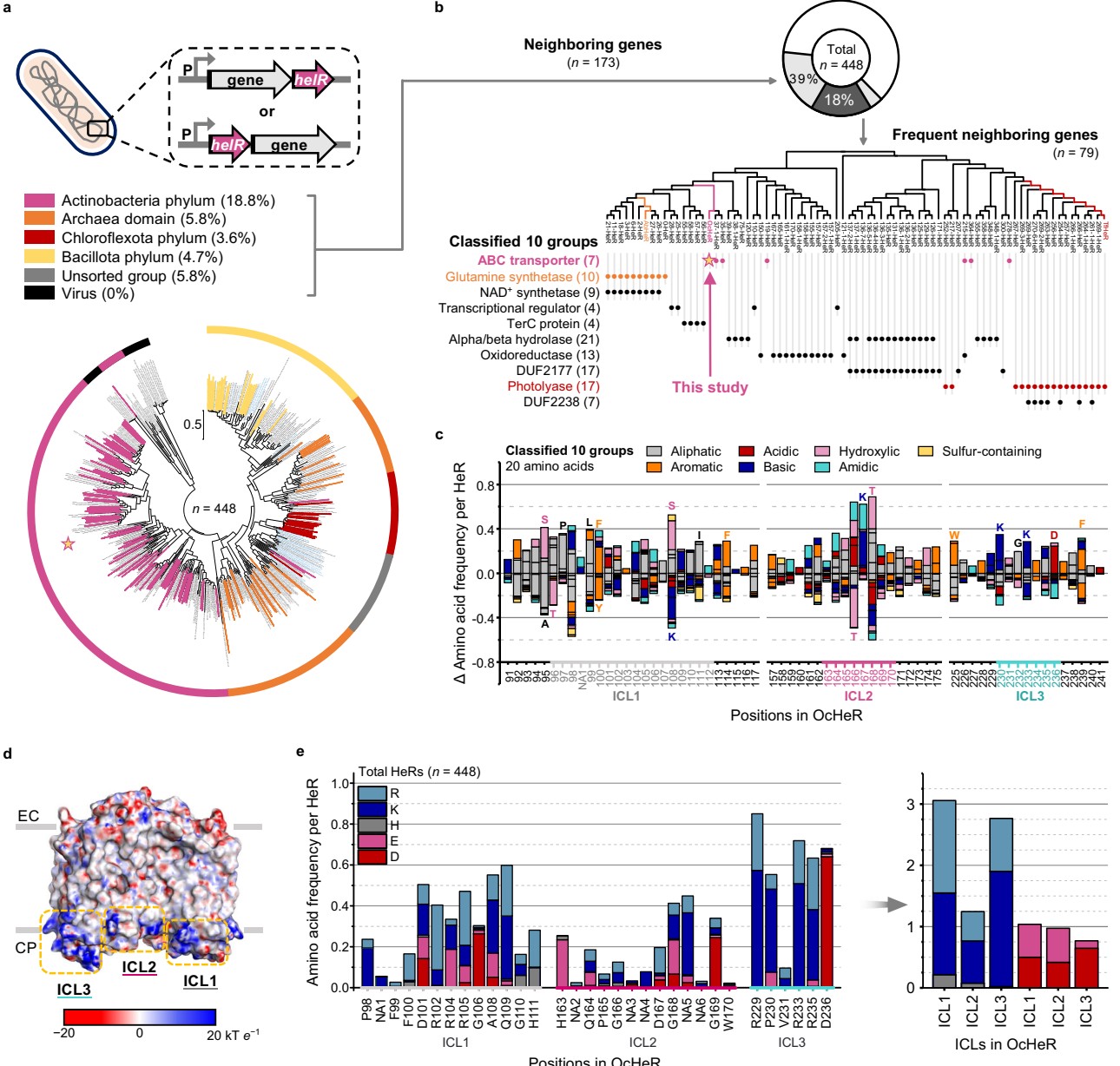

**Fig. 1 | HeR genes flanked by neighboring genes in the same operon.** Phylogenetic trees between the HeRs (total: 448 heliorhodopsin sequences) in organisms (**a**) and HeRs (79 heliorhodopsin sequences) (**b**). **a** Each group of organisms is indicated by a thin-colored donut curve. HeR roots in which at least two genes (*helR* with adjacent genes in the same operon) are predicted to be transcribed by a single promoter and are indicated by the same color belonging to the groups. The percentages of *helR* with adjacent genes per total *helR* are labeled next to each group. **b** Total *helR* (*n* = 448), genes adjacent to *helR* (*n* = 173), and frequent neighboring genes (*n* = 79) are indicated by white, gray, and dark gray pie donut curves. *helR* and adjacent genes in the same operon were analyzed. The adjacent genes were classified into ten groups. Each group is indicated by different colored texts and circles.

**c**, **e** The amino acids of each HeR at the residue positions in OcHeR are counted and normalized by the number of each total HeR, that is, amino acid frequency per HeR. **c** Difference in amino acid frequency per HeR in the classified ten groups compared to non-co-transcription group (*helR* alone). Residues that increased and decreased by more than 0.19 are marked as upper and lower stack-bars, respectively. **e** Amino acid frequency per HeR in total HeRs (*n* = 448). **d** Blue and red surfaces in the electrostatic potential distribution of OcHeR present positively and negatively charged residues, respectively. Red and blue correspond to potentials of 20 kT $e^{-1}$ and −20 kT $e^{-1}$, respectively. The structure is viewed from membrane angles, and yellow dotted rectangles indicate ICLs. ICL intracellular loop, NA not aligned position, EC extracellular side, CP cytoplasmic side.

with canonical and non-canonical residues (Supplementary Fig. 8b). Overall, OmrDE was identified as a multidrug ABC half-exporter that forms heterodimers.

Next, we investigated the photochemical, biophysical, and biochemical properties of OcHeR and OmrDE. The wavelength of maximum absorbance ($\lambda_{max}$) of OcHeR was 554 nm at neutral pH. From pH titration, the p$K$a value through the spectral red-shift of the $\lambda_{max}$ of OcHeR at acidic pH values was estimated to be 2.9, reflecting that the protonation of counterion (presumably E118) (Supplementary

Fig. 9a, b). The p$K$a value through the deprotonation of the retinal Schiff base linked to K252 of OcHeR at alkaline pH values was estimated to be 11.4 based on the difference in absorbance of the deprotonated form of the retinal Schiff base (Supplementary Fig. 9c, d). Right-side-out (RSO) membrane vesicles containing OcHeR did not exhibit ion-pumping activities (Supplementary Fig. 9e). SDS–PAGE revealed high purity of the corresponding proteins upon co-expression of OmrD and OmrE (Fig. 3c). To investigate the heterodimeric form of OmrDE, we constructed different tagged proteins

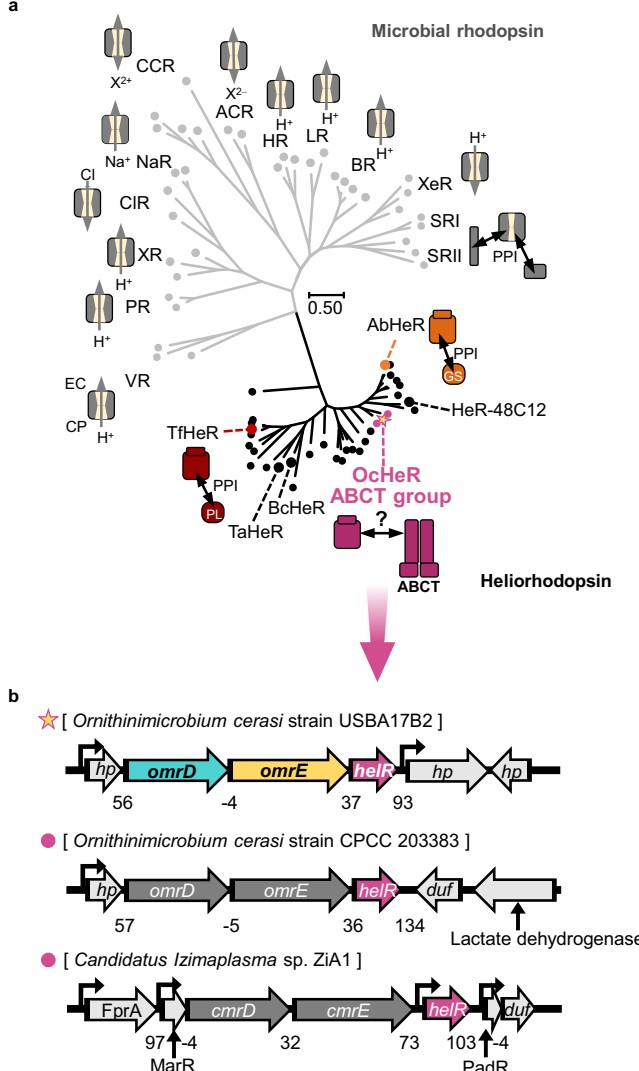

**Fig. 2 | HeRs in the group of ABCT-containing operons. a** Phylogenetic tree between the microbial rhodopsins and HeRs (68 rhodopsin sequences). Each function of rhodopsins is indicated using schematic ion-pumps and -channels in microbial rhodopsins and schematic regulatory function in HeRs. The HeRs belonging to the glutamine synthetase and photolyase groups are represented as orange and dark red circles, respectively. **b** OcHeR and other HeRs in the group of ABCT-containing operons are indicated using pink circles and stars, respectively. The direction of transcription of genes in the operons of the organisms is indicated by arrows. The predicted promoters are indicated using black bent arrows. Nucleotide gaps between the genes are labeled for each gene. EC extracellular side, CP cytoplasmic side, PPI protein–protein interaction, *hp* hypothetical protein, *duf* domain unknown function.

(OmrD::his-tag and OmrE::HA-tag) and performed a His-tag pull-down assay, showing that co-expressed OmrDE formed a heterodimer (Fig. 3d).

The reported multidrug ABCTs transported DNA intercalators–Hoechst 33342 (HO342) and 4',6-diamidino-2-phenylindole (DAPI)[19,20,22,30]. We performed drug-screening tests for OmrDE in the *Escherichia coli* N43 strain (*acrA* mutation) as a minimized cell line for affection drug susceptibilities[31], further testing the minimum inhibitory concentration (MIC) of drugs (chloramphenicol, kanamycin, tetracycline, DAPI, and HO342) in the N43 strain harboring each plasmid [pET21b-empty, -OmrDE, and -OmrDE^m (OmrD WT and OmrE D577Q as the canonical residue mutation associated with ATP binding in NBDs)], termed cell-plasmid. The MIC values of the antibiotics (Cm, Km, and

Tet) for the cells were not strongly elevated by OmrDE and OmrDE^m (Fig. 3e, Supplementary Fig. 10a–f, Supplementary Table 1). Especially, the MIC values of DAPI and HO342 for cell-OmrDE were higher than those for cell-empty. In addition, compared with cell-empty, cell-OmrDE^m was resistant to DAPI and HO342; however, compared with cell-OmrDE, the MIC values were decreased by DAPI and HO342. This suggests that the non-canonical aspartate is responsible, even in the absence of catalytic glutamate (Fig. 3e). Particularly, by comparing the MIC values of DAPI, the MIC values were found that cell-OmrDE was not only 3.9-fold more resistant than cell-empty but also more resistant than that under the same conditions upon HO342 treatment (Fig. 3e).

To evaluate drug transportation capacity, we performed a DAPI translocation assa(y in an inverted membrane vesicle (IMV). The fluorescence intensity for time-tracing by adding ATP can be attributed to two reasons: 1) OmrDE transports DAPI into the IMV from the reaction solution. (2) The fluorescence intensity of DAPI transported inside is relatively reduced owing to interference with the membrane (Supplementary Fig. 11a, b). OmrDE showed a higher transportation capacity compared to the empty vector, OmrDE^m, and OcHeR (Fig. 3f). As endogenous multi-drug ABCTs of *E. coli* also transport DAPI, the fluorescence intensity of the empty vector may be decreased. Interestingly, the capacity of OcHeR was lower than that of the empty vector, suggesting that the embedding positions of the endogenous multi-drug ABCTs in the lipid bilayer are limited by OcHeR overexpression.

Furthermore, we performed ATP hydrolysis of NBDs with increasing concentrations of DAPI via the colorimetric determination of inorganic phosphate ($P_i$), resulting in an inhibitory concentration 50 ($IC_{50}$) value of 47.4 μM (Supplementary Fig. 11c, d). This result, wherein increasing drug concentrations inhibited drug transport, is similar to that reported for multidrug ABCTs[20,30,32]. Moreover, the $V_{max}$ and $K_m$ values of OmrDE and OmrDE^m were determined as enzyme kinetic parameters; the $V_{max}$ and $K_m$ values of OmrDE were 17.6 μM/min and 19.2 mM, respectively, while those of OmrDE^m were 5.1 μM/min and 27.6 mM, respectively (Fig. 3g, Table 1). The $k_{cat}/K_m$ value of OmrDE was 5.6-fold higher than that of OmrDE^m (Table 1), indicating that OmrDE^m showed slight activity; this is consistent with the results of the MIC and DAPI translocation assays. Hence, we suggest the importance of investigating the potential effects of OcHeR for OmrDE (Fig. 3h).

## OcHeR binds to OmrDE
PPIs play various biological roles, including mediating functions, such as environmental sensing, metabolic and signaling enzyme activity, and signal transduction[33]. ITC is an experimental method used to measure thermal change during molecular bindings, resulting in providing binding affinity and various thermodynamic parameters (Supplementary Fig. 12a). To reveal the interaction between OcHeR and OmrDE, we performed ITC analysis and determined dissociation constant ($K_d$), enthalpy (ΔH), entropy (−TΔS), and Gibbs free energy (ΔG) as thermodynamic parameters for binding; we obtained values of 91.6 μM, 48.9 kcal mol⁻¹, −54.3 kcal mol⁻¹, and −5.4 kcal mol⁻¹, respectively (Fig. 4a, Table 2). Furthermore, we observed that OcHeR WT interacted with NBD-truncated OmrDE (OmrD^tcE^tc), suggesting that OcHeR binds to the TMDs (Fig. 4a, Table 2). Specifically, the $K_d$ value of OcHeR WT for OmrD^tcE^tc was 2.4-fold lower than that of OmrDE, indicating that NBDs influenced their binding.

Interestingly, the ΔH value of OcHeR for OmrD^tcE^tc indicated an exothermic reaction, whereas that of OcHeR for OmrDE indicated an endothermic reaction (Table 2). ABCTs undergo conformational changes through ATP binding and hydrolysis in NBDs[15], suggesting that OmrD^tcE^tc forms a different heterodimeric structure from OmrDE. Due to this difference, enthalpic factor contributed from unfavorable (ΔH > 0) to favorable (ΔH < 0) non-covalent interactions, thereby forming new bonds and facilitating the release of energy, as well as influencing the entropy change.

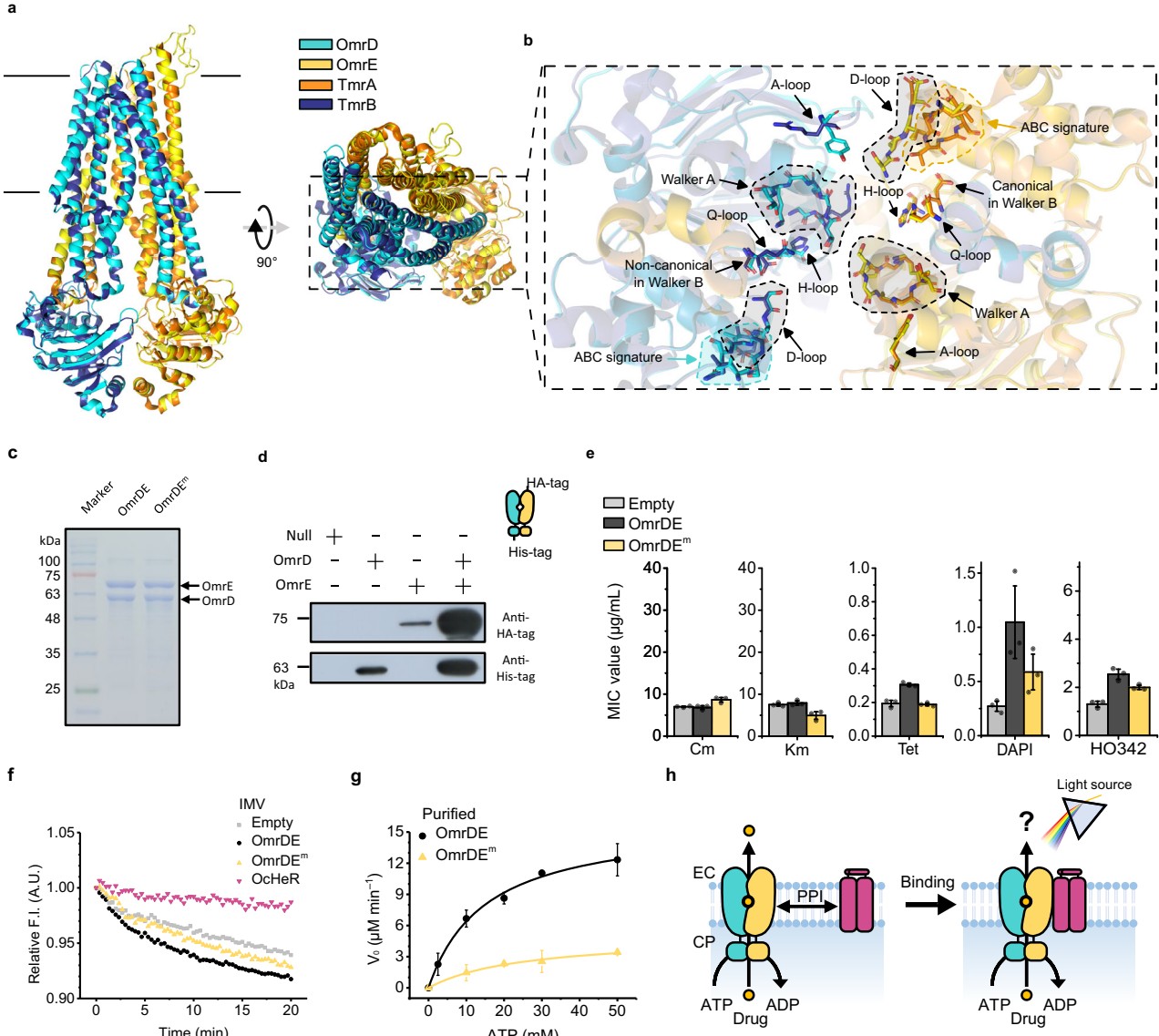

**Fig. 3 | Functional analysis of OmrDE. a, b** Comparison protein structures between OmrDE (predicted by TmrAB as a template, PDB: 6RAF) and TmrAB (only TMDs and NBDs are indicated). The protein structures are colored: OmrD, cyan; OmrE, yellow; TmrA, orange; TmrB, navy. **b** The functional important motifs in OmrDE and TmrAB are indicated by black arrows with labels. Walker A, ABC signature, and D-loop are grouped with irregular shapes indicated as dotted lines. **c** Purified membrane proteins subjected to SDS-PAGE. **d** Western blotting after the His-Tag pull-down assay. The molecular weights of the purified membrane protein containing hexahistidine (His) and trihemagglutinin (HA) tags are calculated using protein sequences: OmrD-His, 62.86 kDa; OmrE-His, 72.09 kDa; OmrE-HA, 75 kDa. **c, d** These experiments were performed twice, and the representative data are shown. **e** MIC tests for *E. coli* harboring plasmids treated with various drugs, as shown in Supplementary Fig. 10a–f, Supplementary Table 1. Measurements were conducted in an independent experimental group ($n = 3$), in which the data were presented as mean value ± SD. Cm, chloramphenicol; Km, kanamycin; Tet, tetracycline; DAPI, 4′,6-diamidino-2-phenylindole; HO342, Hoechst 33342. **f** DAPI translocation in IMVs containing no protein or membrane proteins was initiated by adding Mg-ATP, followed by measuring DAPI fluorescence; measurements were conducted in an independent experimental group ($n = 6$). **g** ATP hydrolysis of the NBDs of purified membrane proteins was performed using colorimetric determination of $P_i$. Non-linear fits were evaluated using the Michaelis–Menten equation, and measurements were conducted in an independent experimental group ($n = 3$), in which the data were presented as mean value ± SD. **h** Suggested model of OcHeR function associated with OmrDE. Membrane proteins are embedded in the membrane. Marker, protein marker; Null, empty vector; CP cytoplasmic side, EC extracellular side.

In addition, we probed the monomer of the OmrDE heterodimer that binds to OcHeR, resulting in OcHeR WT binding to each monomer (OmrD and OmrE; Fig. 4a, Table 2). The $K_d$ value of OcHeR WT for OmrE was 2.6-fold lower than that for OmrDE, unlike that for OmrD, which showed a similar binding affinity. In addition, the ΔG value of OcHeR WT for OmrD indicated a non-spontaneous reaction compared with that for OmrDE and OmrE, which were spontaneous reactions (Table 2). The protein−protein docking simulation analysis revealed that the number of residues in the interfaces between OcHeR and OmrDE was higher for OmrD than those for OmrE (Fig. 4b); however,

OmrD does not spontaneously form a complex (OcHeR−OmrDE). OmrD requires certain conditions to form a complex, suggesting that OmrE may influence the complex formation.

Next, we identified polar interaction residues at the interface between OcHeR and OmrDE on the cytoplasmic side using protein−protein docking simulation (Fig. 4c). The polar interactions were observed in ICL1 and 2: R102 in OcHeR with L136 and V137 in OmrD, and R105 in OcHeR with V18 in OmrD−in ICL1. In addition, G166 in OcHeR interacted with W313 in OmrE−in ICL2. The polar interaction was not observed with residues in ICL3 of OcHeR; however, charged

**Table 1 | Steady-state enzyme kinetic parameters for NBDs in OmrDE with and without OcHeR**

| Omr | OcHeR | Light (532 nm) | $V_{max}$ (µM/min) | $K_m$ (mM, ATP) | $k_{cat}$ (s⁻¹) | $k_{cat}/K_m$ (M⁻¹ S⁻¹) |
|---|---|---|---|---|---|---|
| DE | – | – | 17.6 ± 5 | 19.2 ± 12.3 | 0.293 ± 0.082 | 20.6 ± 9.3 |
| DEᵐ | – | – | 5.1 ± 0.7 | 27.6 ± 10.5 | 0.086 ± 0.011 | 3.7 ± 1.5 |
| DE | WT | – | 18.3 ± 6.9 | 10.4 ± 7 | 0.304 ± 0.115 | 37.7 ± 13.7 |
| DE | WT | + | 20.7 ± 2.2 | 6.3 ± 2 | 0.344 ± 0.037 | 58.4 ± 11.3 |
| DE | R102Q | – | 18.2 ± 0.7 | 14.4 ± 1.8 | 0.303 ± 0.012 | 21.4 ± 2.5 |
| DE | R102Q | + | 19.5 ± 1.7 | 18.1 ± 1.8 | 0.325 ± 0.016 | 18.1 ± 1.7 |

$V_{max}$ and $K_m$ values were calculated using the Michaelis–Menten equation.
$V_O$ values at each ATP concentration are provided in Source Data.
Each of enzyme kinetic parameters of measurements in an independent experimental group ($n = 3$) is provided in Source Data.
Values are presented as mean value ± SD. ATP hydrolysis was conducted at 25 °C.

and aromatic residues in OcHeR were adjacent to OmrD and OmrE. Furthermore, a protein sequence alignment among HeRs in the group of ABCT-containing operons revealed conserved charged or aromatic residues in ICL1–3 (Supplementary Fig. 6b). Specifically, the absence of positively charged residues of reported HeRs in ICL3 has been shown to result in an increased $K_d$ value or no binding (Supplementary Fig. 6b)[12,13]. Considering polar interactions and charged or aromatic residues at the interface between OcHeR and OmrDE via the docking simulation, we constructed the following OcHeR mutants in ICL1–3: R102Q, R104Q, R105Q, D167N, W170A, W225A, R229Q, R233Q, and R235Q. All $\lambda_{max}$ of the mutants were not changed compared to that of WT (Supplementary Fig. 9f).

We performed the ITC analysis of the OcHeR mutants with OmrDE (Fig. 4a, Table 2, Supplementary Fig. 13). The $K_d$ values of mutants in ICL1 and ICL2 were undetectable and non-binding (~704 mM), respectively. In contrast, the $K_d$ values of W225A, R229Q, R233Q, and R235Q in ICL3 were lower than those of the WT. Notably, even one positively charged residue in ICL1 is crucial for binding; moreover, alternation in the positively charged residues at R229 resulted in an 18-fold decrease in the $K_d$ value compared to that of WT (Fig. 4d). Interestingly, the binding affinities according to ICLs showed different results, possibly because HeR forms a symmetrical homodimer. Considering that OcHeR is a homodimer, ICL1 in subunit 1, as well as ICL2 and 3 in subunit 2, were oriented in the same direction, and the non-binding residues of OcHeR for OmrDE were biased toward a specific area (Fig. 4d). Moreover, the simulation showed that each OmrDE monomer binds to each subunit of OcHeR; specifically, OmrD and OmrE adjacent to ICL1 in subunits 1 as well as ICL2 in subunit 2 and ICL3 in subunit 2 (Fig. 4c, d). Owing to the limitations posed by weak binding in OcHeR–OmrDE interactions, perfect saturation was not achieved; however, the $K_d$ values obtained for OcHeR and OmrDE in this study are comparable and relevant because the experimental methods for WT and mutants were identical.

## OcHeR upregulates OmrDE activity

The remaining question was whether the binding of OcHeR to OmrDE affects its transport capacity in the presence of light. Thus, we performed ATP hydrolysis and drug transport in vitro and cell viability assays in vivo in OmrDE with OcHeR WT and OcHeR R102Q. First, the enzyme kinetic parameters of OmrDE with OcHeR were quantified using the ATP hydrolysis assay. In the absence of light, the $V_{max}$ and $K_m$ values of OmrDE with OcHeR WT were similar to, and lower, respectively, than those of OmrDE alone, indicating that the binding of OcHeR to OmrDE increased the substrate-binding affinity. In the presence of light, the $V_{max}$ and $K_m$ values of OmrDE with OcHeR WT were similar to and lower those obtained in the absence of light (Fig. 5a, Table 1). The $k_{cat}/K_m$ values of OmrDE with OcHeR WT in the absence and presence of light were 1.8- and 2.8-fold higher than those of OmrDE alone, respectively. In contrast, the $k_{cat}/K_m$ values of OmrDE bound to OcHeR R102Q in the absence and presence of light were

similar to those of OmrDE alone (Fig. 5b, Table 1), indicating no significant influence unless binding occurred.

Second, we measured the drug transport capacity of OmrDE with OcHeR in IMV. The co-expression of OmrDE with and without OcHeR was observed; however, the expression level of OmrDE in the co-expression system was lower than that in OmrDE expression alone (Supplementary Fig. 14a). Thus, we did not compare the co-expression of those in IMV to OmrDE in IMV. In the presence of light, the capacity of OmrDE with OcHeR WT significantly increased over time compared with that in the absence of light. In contrast, the capacity of OmrDE with OcHeR R102Q did not significantly change, regardless of the presence of light (Fig. 5c). Particularly, the capacity of OmrDE with OcHeR WT in the presence of light significantly differed from that of OmrDE with OcHeR WT and R102Q under other conditions at 15 min. In other words, the capacity of OmrDE with OcHeR WT in the presence of light was 3-fold higher than that in the absence of light. In contrast, the capacity of OmrDE with OcHeR R102Q was not significant (Fig. 5d).

Third, the growth curve of the N43 strain treated with DAPI was recorded. The co-expression of OmrDE and OcHeR in the N43 strain was shown (Supplementary Fig. 14b), and the cell viability assay of cells treated with DAPI harboring plasmids was performed (Supplementary Fig. 10g–m). In the absence of light, cell-DE + WT was resistant compared with cell-DE + R102Q and other cells and was more resistant in the presence of light (Fig. 5e). When the $IC_{50}$ values of cell-WT and cell-empty were compared, OcHeR did not affect cell viability (Fig. 5f). The $IC_{50}$ values of cell-DEᵐ + WT and R102Q were slightly increased by ~1.4 times compared with those of cell-empty; this result was consistent with the slightly lower OmrDEᵐ activity than that of the empty cells (Fig. 3e–g). The $IC_{50}$ value of cell-DE + WT in the absence of light was 2.6-fold higher than that of cell-empty, and that of cell-DE + WT in the presence of light was 3.3-fold higher than that of cell-empty. In contrast, the $IC_{50}$ values of cell-DE + R102Q were independent of light and were lower than those of cell-DE + WT (Fig. 5f).

The photocycle is a series of photochemical reactions of the chromophore and conformational changes in the apoprotein that occur within rhodopsins upon light absorption. In a previous study, the $\lambda_{max}$ and photocycle kinetics of HeR binding to photolyase changed compared with those of photolyase alone[13]. However, the $\lambda_{max}$ values of OcHeR with and without OmrDE were not affected, implying that the PPI between OcHeR and OmrDE does not affect the retinal chromophore (Supplementary Fig. 9g). Additionally, the absorption difference of OcHeR after illumination showed peaks at 554 and 618 nm as the $\lambda_{max}$ of the G and O states, respectively (Supplementary Fig. 9h). The half-life ($t_{1/2}$) values of the G and O states in OcHeR binding to OmrDE were 1.6-fold and 2.2-fold higher than those in OcHeR WT alone, respectively. In contrast, the $t_{1/2}$ values of OcHeR R102Q binding to OmrDE did not substantially differ from those of OcHeR R102Q alone (Fig. 5g, Supplementary Fig. 9i, j).

In a non-essential activation model, similar to reversible mixed inhibition, enzyme activity was activated, resulting in changes in $K_m$

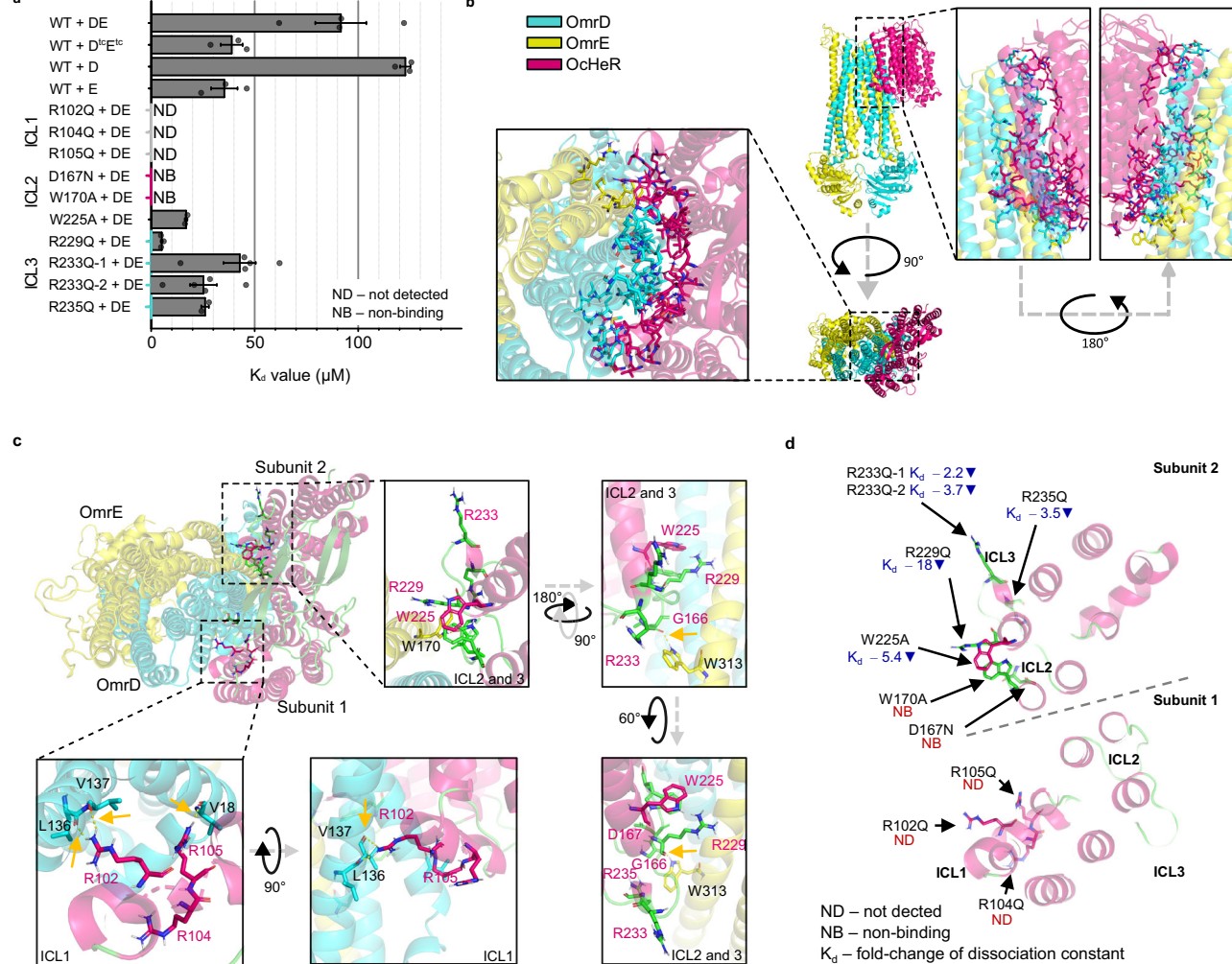

**Fig. 4 | Crucial binding position of OcHeR for OmrDE. a** Binding affinities of OcHeR for OmrD$^{tc}$E$^{tc}$, OmrD, OmrE, and OmrDE from ITC analysis in Table 2 and Supplementary Figs. 12 and 13. The $K_d$ values are estimated using a one site binding model. ITC analysis of OcHeR R233Q for OmrDE is evaluated using a sequential binding site model: R233Q-1, first binding; R233Q-2, second binding. Measurements were conducted in an independent experimental group (*n* = 2 to 5). Data are presented as mean value ± SEM. **b**, **c** Protein–protein docking simulation between modeled OcHeR homodimer based on PDB code: 6SU4 and modeled OmrDE heterodimer based on PDB code: 6RAF. OcHeR, OmrD, and OmrE structures are shown in pink, cyan, and yellow, respectively. **b** Residues at a distance of 5 Å from the interface between OcHeR and OmrDE are shown. **c** OcHeR homodimer is expressed in two subunits (monomer). The distance (2.0–2.2 Å) between the hydrogen bonds of the interacting residues of OcHeR and OmrDE was determined using the polar interaction tool; it is indicated by yellow dotted lines and marked by yellow arrows. The positions of residues in OcHeR and OmrDE are indicated in pink and black text, respectively. **d** Scheme of the crystal structure of OcHeR is shown from the top angle, with each subunit of OcHeR separated by dotted lines. Each mutation position is indicated by text with non-binding and fold-changes of the $K_d$ values of OcHeR mutants for OmrDE compared with that of OcHeR WT for OmrDE. ICL intracellular loop, ND not detected, NB non-binding.

and $V_{max}$. We observed that OmrDE alone could transport drugs and that OcHeR could enhance OmrDE activity via PPI. Moreover, the dynamic energy from the photocycle via PPI may be transferred to OmrDE and affect its conformational changes, enhancing the drug transport capacity (Fig. 5h). Therefore, we suggest that OcHeR functions as a non-essential activator for OmrDE.

## Discussion

Operons typically encode genes in the same functional pathway[34]. Moreover, the intergenic distance frequency distributions of adjacent genes in the same direction show characteristic peaks between −20 and 30 bp[35]. We observed that in the group of glutamine synthetase-containing operons, three genes (glutamine synthetase, NAD⁺ synthetase, and HeR) were encoded in the same direction under a single promoter (Fig. 1b), suggesting that these three genes are associated with nitrogen metabolism in the same functional pathway. In addition, the groups of PHR- and DUF2238-containing operons, as well as the

groups of alpha/beta hydrolase- and DUF2177-containing operons, encoded both genes in the same operon, suggesting that DUFs are involved in the same functional pathways involving these genes. Conversely, the groups of SDR-, TerC-, and MerR family protein-containing operons did not contain other grouped genes in the same operon (Fig. 1b). Recently, a study reported that light-induced viral HeR can transport protons using a whole-cell patch-clamp[14]. In our study, we did not identify neighboring genes flanking *helR* in viruses (Fig. 1a), suggesting that the primary function of viral HeR is its proton-pumping activity rather than protein regulation.

We suggest a compensation between enthalpic (non-covalent) and entropic (hydrophobic) factors in the binding affinity of OcHeR WT and mutants with OmrDE (Fig. 6a; details are provided in the Supplementary Discussion)[36]. OcHeR WT binding to OmrDE shows unfavorable non-covalent and favorable hydrophobic interactions; however, the absence of positively charged residues in ICL1 interferes non-covalent interactions. OcHeR mutants, compared to WT, show

**Table 2 | Thermodynamic parameters of OcHeR for OmrD, OmrE, and OmrDE**

| OcHeR | Omr | $K_d$ (µM) | $\Delta H$ (kcal mol⁻¹) | $-T\Delta S$ (kcal mol⁻¹) | $\Delta G$ (kcal mol⁻¹) |
|---|---|---|---|---|---|
| WT | DE | 91.6 ± 12.3 | 48.9 ± 12.2 | −54.3 ± 12.1 | −5.4 ± 0.1 |
| WT | DᵗᶜEᵗᶜ | 38.9 ± 5.3 | −35.5 ± 2.7 | 29.4 ± 2.5 | −6.1 ± 0.1 |
| WT | D | 122.8 ± 2.5 | NC | NC | NC |
| WT | E | 35.3 ± 6.4 | 35.5 ± 5.2 | −41.6 ± 5.1 | −6.1 ± 0.1 |
| R102Q | DE | ND | ND | ND | ND |
| R104Q | DE | ND | ND | ND | ND |
| R105Q | DE | ND | ND | ND | ND |
| D167N | DE | NB | NB | NB | NB |
| W170A | DE | NB | NB | NB | NB |
| W225A | DE | 16.9 ± 0.6 | 11.8 ± 4.4 | −18.3 ± 4.4 | −6.5 ± 0.02 |
| R229Q | DE | 5.1 ± 0.4 | 8.6 ± 0.7 | −15.8 ± 0.7 | −7.2 ± 0.05 |
| R233Q-1ᵃ | DE | 42.9 ± 7.9 | 5.9 ± 1.1 | −11.9 ± 1 | −6 ± 0.2 |
| R233Q-2ᵃ | DE | 25.3 ± 6.5 | 38.7 ± 10.9 | −45.1 ± 10.8 | −6.4 ± 0.2 |
| R235Q | DE | 26 ± 1.8 | 2.4 ± 0.02 | −8.7 ± 0.07 | −6.3 ± 0.05 |

Measurements were conducted in an independent experimental group ($n$ = 2 to 5) in Supplementary Figs. 12 and 13.

Values are presented as mean value ± SEM. ITC analysis was performed at 25 °C.

Non-binding, not detected, and not calculated are presented by NB, ND, and NC, respectively.

ᵃEach of binding affinities and thermodynamic parameters was calculated using the sequential binding site model of OriginLab.

more favorable non-covalent interactions, indicating the formation of new non-covalent interactions and suggesting a compensatory effect. The comparison between unfavorable non-covalent and favorable hydrophobic interactions, as well as the influence of positively charged residues in ICL1, reveals inconsistency, possibly because of the presence of both attractant and repellent residues in OcHeR. The compensatory effects, mainly observed in R233Q, lead to sequential binding sites, implying stronger attractant residues than repellent ones. Overall, the results suggest the influence of charged or aromatic residues in ICLs on binding affinities, and it can be proposed that the structure of the homodimeric OcHeR may impact the binding affinity of OmrDE. Furthermore, positively charged residues in ICL1 may reduce enthalpy–entropy compensation, facilitating a shift from a non-spontaneous reaction to a spontaneous reaction by changing a positive $\Delta G$ value to a negative $\Delta G$ value.

Based on the thermodynamic parameters, we carefully suggest the sequential binding mechanism (Fig. 6b): (1) The membrane proteins undergo lateral diffusion in the lipid bilayer, leading to their interaction within a range of polar forces (attractant residues) and subsequent mutual influence. (2) The interaction is driven by four forces: attractant 1 (A₁), three positively charged residues in ICL1; attractant 2 (A₂), W170 and D167 in ICL2; repellent 1 (R₁), W225, R229, and R235 in ICL3; repellent 2 (R₂), R233. The first binding is driven by the A₁ and A₂, contributing to a large decrease in the ΔH value (38.9–5.9 kcal mol⁻¹) and favorable ΔG value based on R233Q-1 result. (3) The second binding initiates when A₁ binds completely to OmrD, but A₂ does not completely bind to OmrD and OmrE because the $K_d$ values of W170A and D167N are very weak interactions. Afterward, unfavorable non-covalent interaction through the R₁ and R₂ interferes with the binding; however, the second binding overcomes the R₁ and R₂ by driving the very weak interactions in the A₂ and hydrophobic interactions, which contribute to a large positive −TΔS value and a favorable ΔG value. Moreover, the second binding accelerates the conformational change in the first binding interface of proteins, contributing to lower the $K_d$ values of the second binding with respect to that of the first binding and conformational entropy of residue side chains via negative −TΔS value. (4) Finally, the two proteins bind and stabilize.

The biological relevance of OcHeR and OmrDE in the *O. cerasi* strain USBA17B2 is a topic of interest. The bacterium was isolated in a habitat containing cytotoxic compounds (details are provided in the Supplementary Methods)[37]. Multidrug ABCTs are polyspecific; they can interact with many dissimilar molecules[38]. PPIs can create new binding sites for substrates not originally present in the individual proteins[39,40]. Hence, we suggest three significant biological implications: (1) OmrDE with OcHeR may interact with cytotoxic compounds as biometabolites produced by microorganisms to inhibit the growth of competing microorganisms. (2) OcHeR binding to OmrDE allows *O. cerasi* strain USBA17B2 to resist cytotoxic compounds in the habitat. By transporting new drugs through binding to OcHeR, it can overcome the limitations of OmrDE, which might have been unable to transport certain drugs previously. 3) Fine-tuning (1.8–2.8 folds increase in OmrDE activity) is an advantage in terms of ATP consumption for ATP hydrolysis and energy consumption for OmrDE expression (details are provided in the Supplementary Discussion)[41–44].

A remaining question is why OcHeR regulates OmrDE activity. PPIs can enhance catalytic activity and regulate enzyme activity[45–47]. Most ABC exporters have three conformational changes: Inward-facing (IF) conformation, open cavity on the intracellular side and substrate binding; occluded conformation, close on the intracellular and extracellular sides via ATP binding; Outward-facing (OF) conformation, release of substrate to the extracellular side by ATP hydrolysis[15,48]. We speculate that interaction between OcHeR and OmrDE can affect OmrDE structure. Thus, OcHeR first binds to OmrDE, and subsequently, the substrate binds to the TMDs of OmrDE at IF conformation, enhancing protein stability and stabilizing the substrate binding site. At occluded or OF conformations, dynamic energy via the conformational changes of light-induced OcHeR may accelerate the dynamic structural change of OmrDE. Because the $K_d$ value is in the ~µM range rather than the ~nM range (i.e., antibody), OcHeR may dissociate after substrate release for OmrDE.

Since the determination of the neuromodulatory ability of channelrhodopsin-2, swift advances in the application of optogenetic technology in neurobiology have occurred[49]. The diverse functionality of HeRs has considerable potential for applications in various fields, including optogenetic technology. Daunomycin slows and stops cancer-cell growth via DNA intercalation and is used to treat acute lymphoblastic or myeloblastic leukemias[50,51]. LmrCD and TM278/288 of the multidrug resistance ABCT transport daunomycin, DAPI, and HO342, suggesting that OmrDE transports daunomycin. Membrane vesicles are promising candidates for applications in biomedical technology and nanotechnology[52]. OmrDE may be applied to drug delivery systems, photodynamic therapy, and tumor-targeting effects in membrane vesicles and bioregulation by expression in various cells, similar to optogenetic technology[49,53–55].

In conclusion, our classified ten groups and increased specific residues present a promising starting point for further research, describing function of HeR in microbial drug resistance and identifying a rhodopsin that binds and modulates ABCT. The interaction between OcHeR and OmrDE may be necessary for drug resistance in the *O. cerasi* strain USBA17B2, as it survives in a drug-containing habitat. Our study provides biological insights into the potential for ABCTs and their applications in anticancer therapy and optogenetic technology. However, as many researchers have suggested, HeRs likely have additional functions. Therefore, further research is necessary to elucidate their underlying molecular mechanisms and to explore their functions.

## Methods

### Plasmid preparation

The genome of the *O. cerasi* strain USBA17B2 was collected in the National Center for Biotechnology Information (NCBI, accession number: OBQK01000028) and the Joint Genome Institute (project ID:

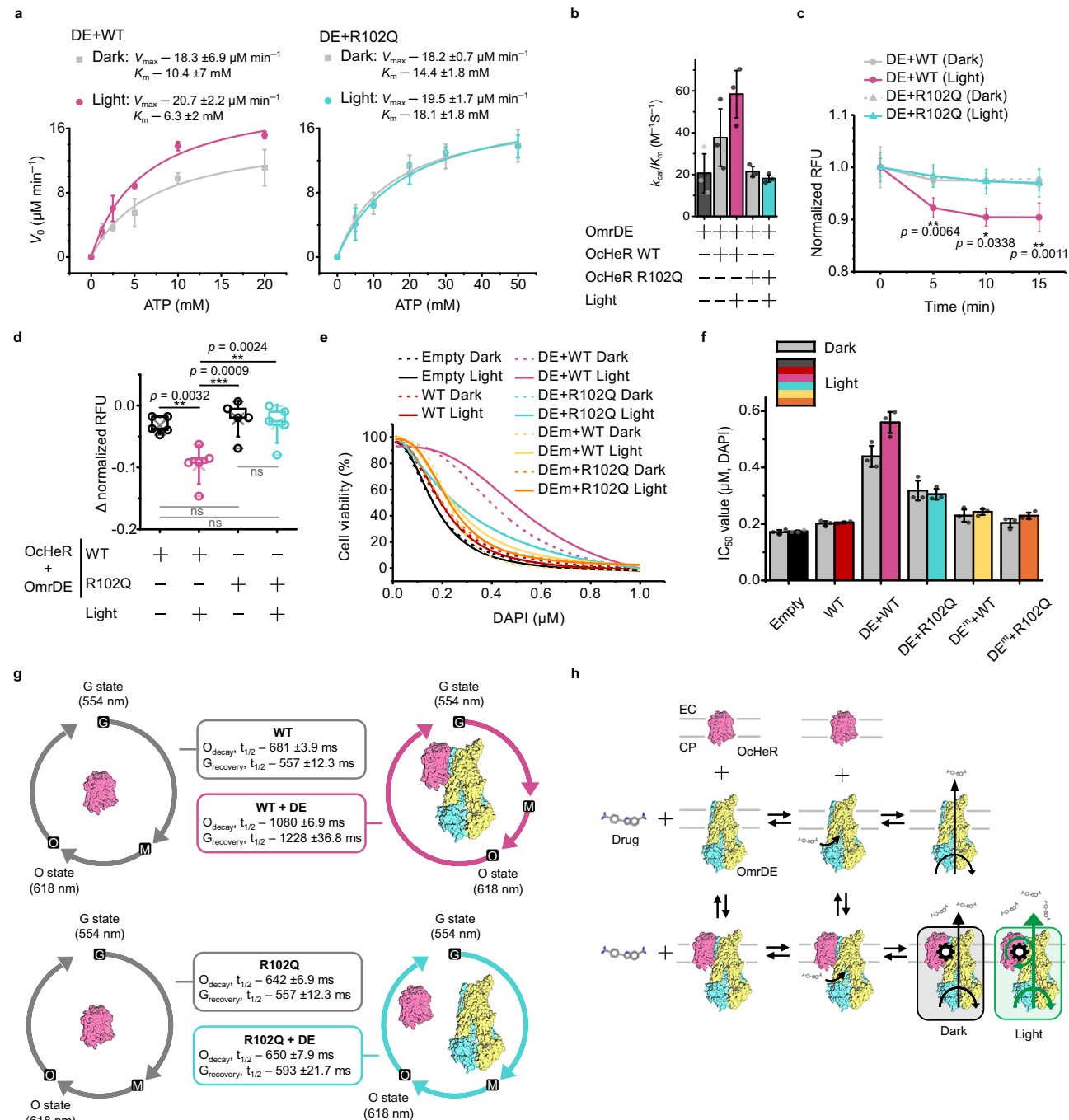

**Fig. 5 | OcHeR-mediated specific light-modulation of OmrDE. a, b** ATP hydrolysis of purified OmrDE with OcHeR using the colorimetric determination of $P_i$. **a** Non-linear fits were evaluated using the Michaelis−Menten equation, and the measurements were conducted in an independent experimental group ($n = 3$). **b** Specificity constants ($k_{cat}/K_m$) are represented as bars. **c, d** Drug translocation of OmrDE and OcHeR co-expressed in IMVs based on the fluorescence intensity in the absence and presence of light. **c** The drug translocation assay was recorded at different time points; the relative fluorescence units (RFUs) were normalized to the initial recording, performed in an independent experimental group ($n = 5$). The significance between the two groups (WT in the absence and presence of light) was analyzed using the two-tailed Student's $t$-test. **d** The delta-normalized RFU means that the data in (**c**) at 15 min were subtracted from the initial recording, which was analyzed using a one-way repeated-measures ANOVA with the post-hoc Dunnett test. The box plots information: minima and maxima, dash marks; centre, cross mark; whisker range, standard deviation; percentile range, 25% (Q1) and 75% (Q3). **e** Non-liner fits were averaged from data in Supplementary Fig. 10h–m, where the x-axis modified logarithmic number to number. Dotted and solid lines indicate the absence and presence of light, respectively. **f** The IC$_{50}$ value for DAPI of data, obtained data in Supplementary Fig. 7h–m, is estimated using the Dose−Response equation in an independent experimental group ($n = 3$). Gray and colored bars indicate the absence and presence of light, respectively. **g** Photocycles of OcHeR without and with OmrDE were recorded at 554 [G (ground) state] and 618 nm (O state); half-life ($t_{1/2}$) values through data analysis were estimated using exponential decay equation as non-linear fitted lines of data in Supplementary Fig. 6i. **h** Schematic model of a non-essential activator. Data information: In (**a–d, f, g**), data are presented as mean value ± SD. In (**c, d**), $p$ values and no significance are indicated with asterisks and labeled "ns", respectively. Exact $p$ values are indicated in the asterisks.

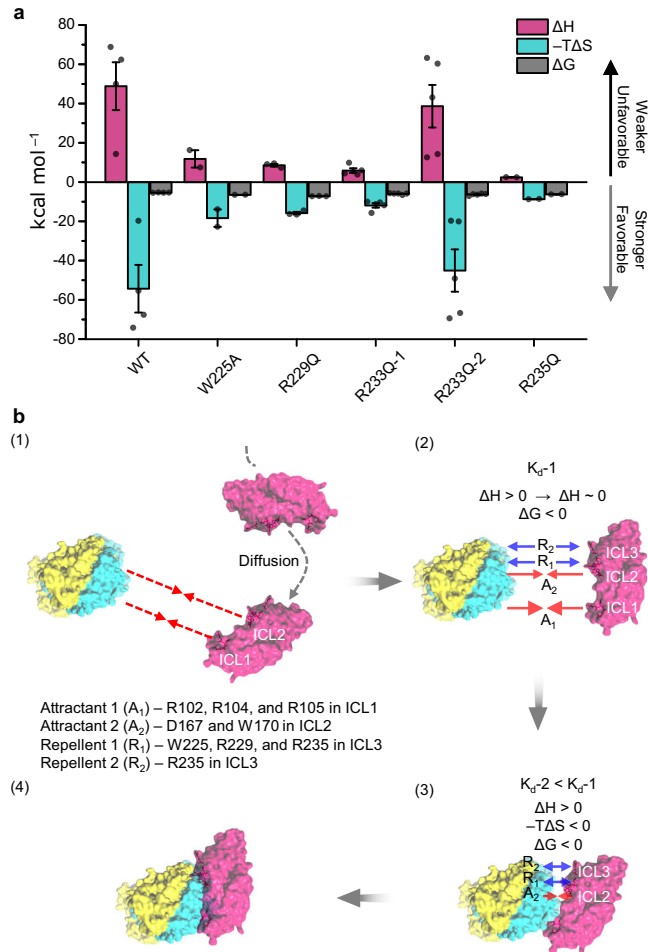

**Fig. 6 | Positive cooperatively sequential binding mechanism. a** Thermodynamic parameters of OcHeR for OmrDE from ITC analysis in Table 2 and Supplementary Figs. 12 and 13. ITC analysis of OcHeR R233Q for OmrDE was performed using a sequential binding site model: R233Q-1, first binding; R233Q-2, second binding. Measurements were conducted in an independent experimental group ($n = 2$ to $5$). data are presented as mean value ± SEM. **b** This mechanism is assumed by OcHeR with IF$_{narrow}$ conformation of OmrDE in the absence of light, which is based on thermodynamic parameters in ITC analysis. Protein structures of OcHeR, OmrD, and OmrE are indicated in pink, cyan, and yellow, respectively. Attractants and repellents are indicated by red and blue arrows, respectively.

1094766). Genes encoding OcHeR (accession number: SOC58301), OmrE (accession number: SOC58303), and OmrD (accession number: SOC58305) were codon-optimized for protein expression in *E. coli* and chemically synthesized (Integrated DNA Technologies, USA). The synthesized genes were introduced into various vectors (details are provided in Supplementary Methods). The oligomers used for preparation and the plasmids constructed in this study are listed in Supplementary Tables 2 and 3.

**Phylogenetic tree and genome analysis**
The phylogenetic tree was analyzed using microbial rhodopsins, OcHeR, reported HeRs (AbHeR, *Actinobacteria bacterium* IMCC26103; TfHeR, *Trichococcus flocculiformis*; HeR-48C12, *Actinobacterium* clone fosmid 48C12; BcHeR, *Bellilinea caldifistulae*; TaHeR, *Thermoplasmatales archaeon* SG8-52-1), and predicted HeRs from data in Supplementary Data 11–18. The predicted HeRs were obtained from a database deposited in the NCBI after analyzing the protein sequences based on the HeR-48C12 protein sequence using protein–protein the Basic Local Alignment Search Tool (BLAST+, Version 2.15.0) with default algorithm parameters[56].

Each of protein sequences was aligned using MUltiple Sequence Comparison by Log-Expectation (MUSCLE) with default algorithm parameters in the European Molecular Biology Laboratory (EMBL)[57]. The aligned sequences were subjected to evolutionary analysis using MEGA 11[58]. Evolutionary analyses were performed using the maximum likelihood method with the JTT matrix-based model[59]. The highest log-likelihoods for the unrooted maximum likelihood were estimated to be −68288.88 (among HeRs from data in Fig. 1a and Supplementary Data 11–18), −12132.14 (among HeRs that flank the neighboring genes in the same operon from data in Fig. 1b and Supplementary Data 19), and −36567.73 (between microbial rhodopsins and HeRs from data in Fig. 3a and Supplementary Data 20). Initial trees were obtained using the neighbor-joining method for a matrix of pairwise distances estimated using the JTT model. The topology with a superior log-likelihood value was subsequently selected. In the operon, the promoters in the nucleotide gaps between *helR* and the neighboring genes were predicted using the phSITE, Softberry, BDGP, and SAPPHIRE promoter tools[60–63].

**Protein expression and purification**
Membrane protein expression and purification procedures were performed as previously described[64]. Protein expression and purification are briefly described and detailed in Supplementary Methods. Membrane proteins were expressed in *E. coli* C43 (DE3) in LB medium by isopropyl β-D-1-thiogalactopyranoside (IPTG) with and without 7 µM all-*trans*-retinal (ATR, Toronto Research Chemicals, Canada). Cells containing membrane proteins were disrupted using sonication and subsequently ultracentrifuged. Membrane fractions were solubilized in buffer S (150 mM NaCl and 50 mM Tris–HCl, pH 7.0) containing 1% (w/w) *n*-dodecyl-β-D-maltopyranoside (DDM, Goldbio, USA). Membrane proteins were purified by an immobilized metal affinity chromatography using Ni$^{2+}$-NTA agarose (Qiagen, USA). Afterward, purified membrane proteins were concentrated by Amicon Ultra MWCO centrifugal filter unit.

**IMV preparation**
IMV, an inside-out membrane vesicle, was prepared using *E. coli* expressing OcHeR and Omr transporters by following a previously reported IMV preparation method for HeR[12]. Membrane protein expression in *E. coli* C43 (DE3) harboring an empty vector (pET21b+), pET21b-DEH, and pET21b-DEH$^m$ was determined according to the method of Omr transporter expression described above. After induction, the cells were harvested with buffer P (5 mM MgSO$_4$ and 50 mM potassium phosphate, pH 7.5), and the *E. coli* pellets were homogenized with 20% (wet weight of pellet/buffer volume) buffer P containing 1 mM dithiothreitol and 1 mM PMSF. The homogenized cells were disrupted using the EmulsiFlex-C3 high-pressure homogenizer (Avestin, Canada) with three passes at 15,000 psi at the Advanced Bio-Interface Core Research Facility, Korea. Unbroken cells and large debris were removed by ultracentrifugation, and IMVs were washed and collected in buffer P by ultracentrifugation. The collected IMVs were centrifuged, and the supernatants were stored in liquid nitrogen. The IMVs were rapidly thawed at 37 °C for 2 min before use.

**Analysis of the photochemical and biophysical properties of OcHeR**
The procedures for estimating the p$K$a value, preparing the RSO membrane vesicles, and conducting light-induced H$^+$ movement assays have been previously described[64]. The OcHeR absorption spectra were measured using a UV–VIS spectrophotometer (Shimadzu UV-2550, Japan) at different pH values after adding HCl and NaOH. The maximum absorbances of rhodopsin and retinal were used as the p$K$a values of the counterion and retinal Schiff bases, respectively. The p$K$a values were estimated using the Henderson–Hasselbalch equation in

OriginPro 9.0 (OriginLab, USA). To measure the light-induced $H^+$ movement, RSO membrane vesicles were resuspended and adjusted to an $OD_{600}$ of 15 ($1.5 \times 10^{10}$ cells/mL) with an unbuffered solution (100 mM NaCl, 10 μM $CaCl_2$, and 10 mM $MgSO_4$), followed by illumination at an intensity of 460 μmol $m^{-2}$ $s^{-1}$ (approximately 100 W/$m^2$) through a short-wave 500 nm cutoff filter (Sigma Koki SCF-50S-44Y, Japan). The assay was performed using RSO membrane vesicles with and without 10 μM protonophores [carbonyl cyanide 3-chlorophenylhydrazone (CCCP)], and the pH was recorded using a pH meter (Horiba pH meter F-71. Japan) with a pH electrode bar.

### His-tag pull-down assay

*E. coli* C43 (DE3) harboring the empty vector, pET21b-OmrD, and pACHA-OmrE, and both pET21b-OmrD and pACHA-OmrE were subjected to the Omr transporter expression and solubilization steps described above. The solubilized membrane proteins were centrifuged at $20,000 \times g$ and 4 °C for 5 min, and supernatants were transferred and quantified using the Bradford assay. The $Ni^{2+}$ NTA agarose resin was added to the supernatants at equal protein concentrations and incubated at 25 °C for 2 h with rocking. The resins were washed five times with buffer SD at $1000 \times g$ for 2 min and resuspended in the Laemmli sample buffer. Samples were vigorously vortexed and incubated at 25 °C for 30 min. The resin in the incubated samples was centrifuged at $15,000 \times g$ for 30 s, followed by SDS–PAGE and western blotting with hexahistidine- and trihemagglutinin (HA)-tagged antibodies. Procedures of SDS–PAGE and western blotting are detailed in the Supplementary Methods.

### Drug resistance test of *E. coli* N43 harboring plasmids

Plasmids (pET21b+, pKA001-DEH, pKA001-DEH$^m$, pKA001-DE$^m$H, and pKA001-DE$^m$H$^m$) were transformed into the *E. coli* N43 strain (*acrA* mutation, National Bioresource Project *E. coli* strain, Japan)[31]. Transformants were selected using 35 μg/mL ampicillin on LB agar plates and further incubated in LB medium containing 35 μg/mL ampicillin at 37 °C and 200 rpm. To induce membrane proteins in the *E. coli* N43 strain harboring the plasmids, the method for Omr transporter expression described above was followed.

Cells harboring the empty vector, OmrDE, and OmrDE$^m$, were incubated to screen for drug transportation. Cells harboring the empty vector, OcHeR WT, OcHeR WT with OmrDE, OcHeR WT with OmrDE$^m$, OcHeR R102Q with OmrDE, and OcHeR R102Q with OmrDE$^m$ were incubated to investigate DAPI transportation. These cells were harvested and washed three times with MOPS minimal medium (40 mM 3-morpholinopropane-1-sulphonic acid, 4 mM tricine, 10 μM $F_2SO_4$, 9.52 mM $NH_4Cl$, 276 μM $K_2SO_4$, 500 nM $CaCl_2$, 528 μM $MgCl_2$, 50 mM NaCl, 1.32 mM $K_2HPO_4$, 3 nM $(NH_4)_6Mo_7O_{24}$, 400 nM $H_3BO_3$, 30 nM $CoCl_2$, 10 nM $CuSO_4$, 80 nM $MnCl_2$, 10 nM $ZnSO_4$ at pH 7.4, and 0.4% D-glucose)[65]. The cells were resuspended in MOPS minimal medium and adjusted to an $OD_{700}$ of 0.1 ($1.0 \times 10^7$ cells/mL); 200 μL of cells were transferred into a 96-well microplate, with each well containing 5 μL of the different concentrations of the drugs (chloramphenicol, tetracycline, kanamycin, Hoechst 33342, and DAPI). The plates were subsequently incubated at 25 °C and 950 rpm (FinePcr, Korea) in the absence and presence of light (specific green laser, 532 nm) at 25 μmol $m^{-2}$ $s^{-1}$. Further $OD_{700}$ measurements of *E. coli* were performed using the 2300 EnSpire Multimode Plate Reader (PerkinElmer, USA). The $MIC_{80}$ value, the lowest concentration of a drug that inhibits cell growth by up to 80% compared with cells without the drug, was determined. Dose–response analysis was performed using the DoseResp function (OriginPro 9.0), which follows Eq. 1a:

$$y = A1 + \frac{A2 - A1}{1 + 10^{(Logx0 - x)p}} \tag{1a}$$

Derived parameters: *A1*, bottom asymptote; *A2*, top asymptote; *Logx0*, center; and *p*, hill slope. $IC_{50}$ values were determined using Eq. 1b:

$$IC_{50} = 10^{Logx0} \tag{1b}$$

### PPI using ITC analysis

ITC analysis of HeR and proteins was performed according to previously reported methods, with slight modifications[12,13]. Purified OcHeR, as well as OmrD, E, DE, and D$^{tc}$E$^{tc}$, were completely exchanged with buffer SD using the Amicon Ultra-4 10,000 and 50,000 MWCO centrifugal filter units, respectively. Purified OmrD$^{tc}$E$^{tc}$ was completely exchanged with buffer SD using the Amicon Ultra-4 30,000 MWCO centrifugal filter units. After the exchange, the concentrated proteins were quantified using the Bradford assay. In the PPI, OcHeR was continuously injected into OmrD, E, and DE at 25 °C for 500 rpm using the MicroCal ITC200 injector (Malvern Panalytical, United Kingdom) at the Advanced Bio-Interface Core Research Facility, Korea. ITC data for the thermodynamic parameters were evaluated using Origin 7.0 SR4 (Origin Lab, USA) with one and sequential binding site models.

### ATP hydrolysis assay

The ATPase activity of OmrDE or a complex protein (OmrDE binding to OcHeR) was measured based on released $P_i$, which was measured via the colorimetric determination of $P_i$ through phosphomolybdate in acetone[66]. First, the purified OcHeR and OmrDE, quantified by the Bradford assay, were incubated at a protein-to-protein molar ratio of 1:1 at 25 °C for 30 min with rocking in the dark. After binding, the drug was added, and the mixture was gently mixed. An Mg-ATP solution (the ratio of $MgSO_4$ and ATP was 1.5:1) was added to activate ATPase. The reaction was performed at 25 °C in the absence and presence of light (specific green laser, 532 nm) at 35 μmol $m^{-2}$ $s^{-1}$ (approximately 7.61 W/$m^{-2}$). The final concentration of the reactants was 150 mM NaCl, 50 mM Tris–HCl at pH 7.0, 0.02% (w/v) DDM, 1 μM complex protein, and different concentrations of the drug and Mg-ATP solution. The reaction was stopped by adding ice-cold $Na_2$-EDTA to a final concentration of 170 mM. An eightfold stopped sample volume of freshly prepared AAM solution (2 parts acetone, 1 part 10 mM ammonium molybdate, and 1 part 5 N $H_2SO_4$) was added; the sample was developed for 5 min. Then, a 0.1-fold stopped sample volume of 1 M citric acid was added to terminate the development process. Finally, $P_i$ concentration was measured at a wavelength of 355 nm using the 2300 EnSpire Multimode Plate Reader. With OmrDE alone, the reaction was performed using the abovementioned procedure (colorimetric determination of $P_i$) without OcHeR binding. The $P_i$ concentrations were calculated using a $P_i$ standard curve, prepared using the procedure described above with different concentrations of potassium phosphate buffer at pH 7.5 without proteins.

### Drug translocation assay in IMVs

Previously reported methods of drug transport via ABCT were followed with slight modifications[19,20,22]. IMVs containing the proteins (OmrDE and OcHeR) were diluted with buffer P, and DAPI was subsequently added. After stabilization of DAPI translocation, OmrDE activity was carried out by adding Mg-ATP solution at 25 °C in the absence and presence of light (specific green laser, 532 nm) at 40 μmol $m^{-2}$ $s^{-1}$ (approximately 8.7 W/$m^{-2}$). The reactant compositions were as follows: buffer P, IMVs ($OD_{650} = 0.1$), 1 μM DAPI, and Mg-ATP solution (15 mM $MgSO_4$ and 10 mM ATP). The fluorescence intensity via DAPI translocation was measured using the 2300 EnSpire Multimode Plate Reader at excitation and emission wavelengths of 360 and 490 nm, respectively.

## Time-resolved UV–visible measurement

For protein binding or non-binding, the purified OcHeR with and without OmrDE was incubated at 25 °C for 1 h, with a 1:1 molar ratio of proteins. The samples were illuminated with white light at 460 μmol m$^{-2}$s$^{-1}$ (approximately 100 W/m$^{-2}$). After illumination, differences in the absorbance of OcHeR in the dark and under light were fitted and peaked at two wavelengths. The two wavelengths of the samples were monitored every 20 ms using the Scinco UV–VIS spectrophotometer (S-3100, Korea) after illumination to excite rhodopsin for 10 s. The data were fitted using ExpDec1 function in OriginPro 9.0, which follows Eq. 2a:

$$y = y_0 + A_1 e^{-x/t_1} \tag{2a}$$

Derived parameters: $A_1$, offset; $A$, amplitude; and $t$, time constant. Half-life ($t_{1/2}$) value was determined using Eq. 2b:

$$t_{1/2} = t_1 \times \ln 2 \tag{2b}$$

## Protein crystal structure analysis

The protein sequences of OcHeR, OmrD, and OmrE were analyzed using the Swiss Model with the available protein crystal structure HeR-48C12 (PDB code: 6SU4) for OcHeR and TmrAB inward-facing conformation (PDB code: 6RAF) for heterodimeric OmrDE[26–28,67]. The predicted crystal structures were analyzed using the PyMOL program (PyMOL Molecular Graphics System, Version 2.52, Schrödinger, LLC). To analyze electrostatic potential distribution, the predicted crystal structure of OcHeR was determined using CHARMM-GUI 3.8[68].

Protein–protein docking simulation between the OcHeR homodimer and OmrDE heterodimer was conducted using the ClusPro 2.0 server[69]. Three computational steps were performed using the ClusPro 2.0 server. Rigid body docking was used to sample billions of conformations using PIPER, which calculates the docked conformation energies in grids using a fast Fourier transform correlation approach. Root-mean-square deviation was generated to determine the largest clusters. Finally, the samples were refined using CHARMM minimization. The docking server-generated various models of balanced, electrostatic-favored, hydrophobic-favored, and van der Waals and electronic sets for energy coefficients with weighted scores.

## Statistical analysis

Statistical analyses were performed using the OriginPro 9.0. The indicated values represent the mean ± SD (standard deviation) and were analyzed using the independent two-tailed Student's $t$-test and one-way repeated-measures ANOVA using the post-hoc Dunnett test. The following $P$-values indicate statistical significance: *$p < 0.05$, **$p < 0.01$, and ***$p < 0.001$. $N$-values are indicated within figure legend and refer to biological replicates. The statistical analyses applied in each experiment are described in the figure legends.

## Reporting summary

Further information on research design is available in the Nature Portfolio Reporting Summary linked to this article.

## Data availability

The data that support this study are available from the corresponding authors upon request. The genome of the *O. cerasi* strain USBA17B2 is collected in the NCBI under accession number OBQK01000028 and the Joint Genome Institute under project ID 1094766. Genes encoding protein used to recombinant protein in this study are available in NCBI under accession numbers SOC58301 (OcHeR), SOC58303 (OmrE), and SOC58305 (OmrD). The template protein crystal structures for predicting the protein crystal structure of OcHeR and OmrDE are available in Protein Data Bank under PDB codes 6SU4 (HeR-48C12) and 6RAF

(TmrAB inward-facing conformation). All data used in this study are available in "Methods", Supplementary information, Supplementary data, and Source data. Source data are provided with this paper.

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

## Acknowledgements

We would like to Prof. Jung's Protein Biochemistry Lab members for their effort, support, and helpful discussion during this study. This research was supported by Basic Science Research Program through the National Research Foundation of Korea (NRF) grant funded by the Ministry of Science and ICT (MSIT, Grant No: RS-2023-0020863). This research was supported by Basic Science Institute (National research Facilities and Equipment Center) grant funded by the MSIT (Grant No: 2020R1A6C101A192).

## Author contributions

S-G.C., J-H.K., and K-H.J. developed and supervised this work. S-G.C. and J-H.K. contributed equally to this work. S-G.C. performed genome analysis, protein characterization, crystal structure analysis, pull-down assay, MIC test, ITC analysis, and in vitro assays. J-H.K. investigated ABC transporters and performed growth tests in vivo. J-e.L supported the results of ITC analysis. I-J.C. prepared the plasmids of HeR mutant. M.S. prepared plasmid related to the pull-down assay. K.C., J-g.S., and K-W.K. discussed the concept and supported plasmid preparation. S-G.C. prepared and revised the manuscript. J-H.K. and K-H.J. discussed and revised the manuscript. All authors have approved the final version of the manuscript.

## Competing interests

The authors declare no competing interests.
