## [Peer Review File · Nature Communications]

Heliorhodopsin-mediated light-modulation of ABC transporterReviewer #1 (Remarks to the Author):

The manuscript by Cho et al. entitled "Novel insights into photoreceptor-mediated specific light modulation of ABC transporter" reports a previously unknown function of heliorhodopsins (HeRs) in microbial drug resistance and thus identifies a photoreceptor that binds and modulates ATP-binding cassette transporters (ABCTs). HeRs are a recently discovered, vast family of retinal-binding proteins, but their cellular functions remain largely unknown. This study is potentially a valuable contribution to HeR research. I do not find any serious problems in this manuscript, but it requires a major revision to make the Authors' findings more comprehensible for the readers. Below I list some of my suggestions, but ideally, the entire manuscript should be edited using professional language services.

Page 2, line 13: "Heliorhodopsin (HeR), a photoreceptor, has been hypothesized to have widespread functions"

As the Authors themselves show in Fig. 1, heliorhodopsins form a large family of proteins, so using the singular form of the word "heliorhodopsin" when referring to the entire family is inappropriate. Please change it to the plural form, "heliorhodopsins", in this sentence and in the entire manuscript when you use it to describe the entire family. Also, it is unclear why the Authors refer to heliorhodopsin as a photoreceptor. It is well known that not all retinal-binding proteins act as photoreceptors (some are involved in energy conversion), and the cellular roles of most heliorhodopsins are not yet known. Please delete the words "a photoreceptor".

Page 2, lines 14-15: "Recently, the functions of HeR have been suggested; however, the functions remain largely unknown"

This sentence is also very awkward. The Authors apparently mean that hypothetical functions, suggested for heliorhodopsins, have not yet been verified experimentally. If this is what you mean, please change the sentence accordingly.

Page 2, line 18: "we classified adjacent genes flanking heR (HeR opsin-encoding)"

The letter "R" in "HeR" means rhodopsin, therefore, you should not repeat the word "opsin". "Adjacent" and "flanking" are synonyms, so one of them will be enough. I suggest to change the quoted phrase to "we classified genes flanking the HeR-encoding genes" for clarity.

Page 3, lines 35-37: "Rhodopsin, a photoreceptor, is a light-driven seven-transmembrane protein with a retinal molecule that binds to an internal pocket and is found in both prokaryotic and eukaryotic organisms"

See a comment above regarding using the singular; it is even more relevant for the word "rhodopsin" than for "heliorhodopsin". Please delete the word "photoreceptor", and replace "molecule" with "chromophore", as bound retinal becomes part of rhodopsin molecule. Also, the retinal chromophore is covalently bound not to the entire pocket, but to a specific Lys residue. Please add viruses to the list of rhodopsin sources.

Page 3, lines 44-45: "Unlike type-I and -II rhodopsins, the topology of HeR is invertedly embedded"
Please change to: "topology of HeRs is inverted, as compared to that of type-I and -II rhodopsins"

Page 3, lines : Particularly, bacterial HeRs, similar to sensory rhodopsin (SR), lack ion pumping activity and have a long-lived photoactivated state".

"Similar" here is confusing, and "particularly" has no sense. Please change to: "Bacterial HeRs lack ion pumping activity and have a long-lived photoactivated state, as do sensory rhodopsins (SRs) that belong to type I".

Page 3, line 52: "HeR functions remained largely unknown"

Please replace with: "they have not yet been tested experimentally".

Page 3, line 58: "A closely adjacent gene"

A gene cannot be "closely adjacent": it is either "adjacent" or "closely located".

Page 4, lines 76-77: "Crucial interaction positions between OcHeR and OmrDE were performed"
Positions cannot be performed; please rephrase to clarify what you mean.

Page 4, line 90-91: "suggesting that these patterns were identified only in prokaryotes"
Please change to "suggesting that these patterns occur only in prokaryotes"

Page 4, line 92: "Considering promoters, terminators, and closely neighboring genes..."
Do you mean "...and coding regions"?

Page 4, lines 92-93: "we classified the HeRs capable of transcribing helR"
This sentence makes no sense at all. The genes, including helR genes, are transcribed by RNA polymerases, not by heliorodopsin proteins.

Page 4, line 98: "we investigate functional key residues that interacts to neighboring gene proteins"
Please change to: "we identified key residues responsible for interaction with proteins encoded by the neighboring genes"

Page 4, lines 104-105: "Particularly, the number of amino acids in hydroxylic, aliphatic, aromatic, acidic, and basic amino acids were increased"
Please change to: "Particularly, the number of amino acids in hydroxylic, aliphatic, aromatic, acidic, and basic groups were increased"

Page 5, line 108: "are more potential to have protein-protein interactions (PPIs)"
Please change to "are more likely to have protein-protein interactions (PPIs) than" and add than what.

Page 5, lines 109-110: "we aimed ABCT group because HeR may interact with ABCT as membrane protein similar to SR interacting with membrane proteins"
This is highly confusing. I understand that by the "ABCT group" the Authors mean such helR genes that have genes encoding ABCT as their neighbors, but the reader may think that the Authors mean ABCT genes themselves. Therefore, I suggest to use here and elsewhere something like "a group of ABCT-containing operons" instead of just "ABCT group". Also, please change "aimed" to "focused on".

Page 5, lines 112-113: "OcHeR exhibited protein sequence similarity with reported HeRs rather than with microbial rhodopsins"
This information is redundant: if OcHeR would be homologous to microbial rhodopsin, it would not be classified as heliorhodopsin! Please delete this sentence.

Page 5, line 125: "functional critical motifs"
Please change to "functionally important motifs"

Page 5, lines 129-130: "the catalytic glutamate next to the Walker B motif in OmrE, the ATP-binding site in the NBD, remained canonical"
Please change to: "the canonical catalytic glutamate next to the Walker B motif in OmrE, the ATP-binding site in the NBD, is conserved"

Page 5, lines 131-132: "phylogenetic tree among multidrug ABC exporters"
Please change to "phylogenetic tree of multidrug ABC exporters"

Page 5, line 139: "pKa values of its counterion and retinal Schiff base were 2.883 and 11.385"
I doubt that it is reasonable to provide three decimal places for the constants; please reduce to one, i.e., 2.9 and 11.4. Also, please briefly explain why you attributed these constants to the counterion and retinal Schiff base, respectively.

Page 5, lines 141-142: "The co-expression of OmrD and OmrE revealed high purity"
Please change to: "Western blotting revealed high purity of the corresponding proteins upon co-expression of OmrD and OmrE"

Page 6, line 147: "We investigated drug-screening"
Carried out drug screening?

Page 6, lines 150-152: "in the N43 strain harboring plasmids, that is, cell-plasmid [empty, OmrDE, and OmrD WT with OmrE D577Q as the canonical residue mutation associated with ATP binding in NBDs (OmrDEm)]"

What is "cell-plasmid"? Also, it is unclear whether the N43 strain carried all three plasmids simultaneously, or these plasmids were tested in parallel. Please explain.

Page 6, line 153: "were not significantly inhibited"
Please change to "were not strongly reduced". Please note that the word "significantly" means the results of statistical hypothesis testing, which has not been done in this case.

Page 6, line 159: "not only 3.85-fold more resistant"
Again, please reduce the number of decimal places to one, i.e., 3.9-fold. Please also do the same for other numeric estimates in the text.

Page 7, line 202: "accelerated compared with diffusion".
Unclear, how charge interaction can be accelerated, and why should it be compared with diffusion?
Please explain, and please also explain why you consider only long-distance charge interactions.

Page 7, line 217: "HeRs showed a greater conservative positively charged residues"
Please change to "HeRs showed a greater conservation of positively charged residues", and also change "implying" to "suggesting" in the next line.

Page 9, line 290: "were similar to and lower than those of OmrDE alone"
Please change to ""were similar to, and lower, respectively, than those of OmrDE alone"

Page 10, line 322: "The photocycle is a photochemical reaction of conformational changes and molecular events"
Please change to "The photocycle is a series of photochemical reactions of the chromophore and conformational changes in the apoprotein"

Page 10, line 345: "encoded in the same direction as the single promoter"
Please change to "encoded in the same direction under a single promoter"

Page 12, lines 398-399: "potential applications to the target are indirect because of its functions as a channel, pump, and enzyme-fused protein"

Unclear, what the Authors mean here. Each microbial rhodopsin used in optogenetics has its unique mode of action, and those that exhibit, e.g., channel activity, do not act as photoregulated enzymes.

Reviewer #2 (Remarks to the Author):

Heliorhodopsin (HeR) is a novel member of rhodopsin family. The discovery of this member, which is widely distributed in microorganisms and lower eukaryotes, came as a great surprise to many researchers. Despite its widespread distribution, the physiological role of HeR has not been fully clarified. In this study, Cho et al. reported that some HeRs make complex with the multidrug efflux

transporters and can elevate the drug transport activities under illumination. This finding is very interesting and deserves to be widely known. However, for many readers, it is not so easy to understand the content of current manuscript. Thus, I'd like to request the authors to consider the following points:

1) Structure of the manuscript

The most important data should be 1) increased drug transport activity of a multidrug efflux transporter OmrDE by the complex formation with a heliorhodopsin named OcHeR and 2) further increase of the OmrDE activity under illumination. However, these data are briefly described after a lengthy description about other data, such as a) the amino acid sequence comparison among many HeRs, b) predicted structure of the multidrug transporter, and c) the ITC data to explore complex formation and its contact points, and so on. To understand these "other data", the readers need to refer so many supporting figures and experimental procedures. I think this manuscript could be revised to something simpler and more readable.

2) Data showing the activity of OmrDE

Since these data are the hearts of this study, the authors should explain their details.

For Fig. 2d, the authors should explain the following three points:

(a) why the fluorescence intensity decreases when DAPI is transported into the inverted membrane vesicles (IMVs), (b) why the fluorescence decrease also occurs in the "Empty" IMV, whose membrane does not contain OmrDE, and (c) why "Empty" IMV has lower fluorescence intensity compared to "OcHeR" IMV.

For Fig. 2e and Fig. 6a, the authors should show the raw data used to calculate the V_0 values at each ATP concentration.

For Fig. 6e, I could not understand why distinct difference between "DE+WT Dark" and "DE+WT Light" was appeared in the cell viability. These two traces should have been calculated from the data in Fig. S7j, which indicates little difference in the cell viability under Dark and Light conditions.

3) Comparison of amino acid sequences of heliorhodopsins

Significance of Fig. 2c, Fig. 4c, and other similar figures are not clear. Moreover, it is probably difficult for most readers to understand how these figures were prepared. Based on these data, the authors determined the amino acids to be mutated for ITC experiments and activity measurements of OmrDE. Instead of this method, I think that the authors should simply use the results of docking simulation of OmrDE and OcHeR (Fig. 3f, Fig. 5a) to determine the residues to be mutated.

4) ITC experiments

Because of the weak binding, saturation of binding was not observed in the most data (Fig. 3b, Fig. S10). Thus, it is questionable how correct the values of thermodynamic parameters determined from these data are. As noted above, the most important result in this paper is that binding of OcHeR alters the activity of OmrDE (Fig. 6a-f). Therefore, the ITC results are not essential. I'd like the authors to reconsider whether they can have a credible discussion based on the ITC data.

5) Swiss-model structure of OmrDE

To determine the type of ABC transporter to which OmrDE belongs, the authors compared the Swiss-model structure of OmrDE with the crystal structures of ABC transporters already reported (Fig. S4). The Swiss-model structure largely depends on the template structure. Probably, the authors chose the TmrAB structure as the template based on its high amino acid sequence similarity to OmrDE (Fig. S5). Thus, without comparing the structures, the authors were able to determine the type of OmrDE, which should belong to the same group as TmrAB.

Author's point by point response to the reviewers' comments

Manuscript ID: NCOMMS-23-44201A

Title: Novel insights into particular rhodopsin-mediated specific light-modulation of ABC transporter

Our point-by-point response to reviewers' comment are detailed as follows.

Responses are presented subsequent to the **Comments**. And the revised texts on our manuscript are **highlighted**.

Reviewer #1 (Remarks to the Author):

The manuscript by Cho et al. entitled "Novel insights into photoreceptor-mediated specific light modulation of ABC transporter" reports a previously unknown function of heliorhodopsins (HeRs) in microbial drug resistance and thus identifies a photoreceptor that binds and modulates ATP-binding cassette transporters (ABCs). HeRs are a recently discovered, vast family of retinal-binding proteins, but their cellular functions remain largely unknown. This study is potentially a valuable contribution to HeR research. I do not find any serious problems in this manuscript, but it requires a major revision to make the Authors' findings more comprehensible for the readers. Below I list some of my suggestions, but ideally, the entire manuscript should be edited using professional language services.

Response: We greatly appreciate the reviewer's detailed reading and believe that the manuscript and presentations were significantly strengthened owing to the changes made in response to the reviewer's comments.

Comments from Reviewer #1

Comment 1: Page 2, line 13: "Heliorhodopsin (HeR), a photoreceptor, has been hypothesized to have widespread functions" As the Authors themselves show in Fig. 1, heliorhodopsins form a large family of proteins, so using the singular form of the word "heliorhodopsin" when referring to the entire family is inappropriate. Please change it to the plural form, "heliorhodopsins", in this sentence and in the entire manuscript when you use it to describe the entire family. Also, it is unclear why the Authors refer to heliorhodopsin as a photoreceptor. It is well known that not all retinal-binding proteins act as photoreceptors (some are involved in energy conversion), and the cellular roles of most heliorhodopsins are not yet known. Please delete the words "a photoreceptor".

Response: Thank you for the query and suggestion. We agree with your comment and have accordingly revised "photoreceptor" throughout the manuscript. Furthermore, we have modified the title of the manuscript "Novel insights into photoreceptor-mediated specific light-modulation of ABC transporter" to "Novel insights into particular rhodopsin-mediated specific light-modulation of ABC transporter".

Comment 2: Page 2, lines 14-15: "Recently, the functions of HeR have been suggested; however, the functions remain largely unknown"

This sentence is also very awkward. The Authors apparently mean that hypothetical functions, suggested for heliorhodopsins, have not yet been verified experimentally. If this is what you mean, please change the sentence accordingly.

Response: Thank you for pointing this out. We agree with this comment. Since the functions (enzyme activity regulations and proton transport) were experimentally revealed, we have revised the sentence as follows:

"Recently, the functions for few HeRs have been revealed; however, the hypothetical functions remain largely unknown" on page 2, lines 13-15.

Cho, S.-G. *et al.* Heliorhodopsin binds and regulates glutamine synthetase activity. *PLOS Biology* **20**, e3001817 (2022).

Shim, J. *et al.* Heliorhodopsin Helps Photolyase to Enhance the DNA Repair Capacity. *Microbiology Spectrum* **10**,

e02215-22 (2022).

Hososhima, S. *et al.* Proton-transporting heliorhodopsins from marine giant viruses. *eLife* **11**, e78416 (2022).

Comment 3: Page 2, line 18: “we classified adjacent genes flanking *heIR* (*HeR* opsin-encoding)”
The letter “R” in “*HeR*” means rhodopsin, therefore, you should not repeat the word “opsin”. “Adjacent” and “flanking” are synonyms, so one of them will be enough. I suggest to change the quoted phrase to “we classified genes flanking the *HeR*-encoding genes” for clarity.

Response: Thank you for pointing this out. We agree and have revised this according to your suggestion.

Comment 4: Page 3, lines 35-37: “Rhodopsin, a photoreceptor, is a light-driven seven-transmembrane protein with a retinal molecule that binds to an internal pocket and is found in both prokaryotic and eukaryotic organisms”.
See a comment above regarding using the singular; it is even more relevant for the word “rhodopsin” than for “heliorhodopsin”. Please delete the word “photoreceptor”, and replace “molecule” with “chromophore”, as bound retinal becomes part of rhodopsin molecule. Also, the retinal chromophore is covalently bound not to the entire pocket, but to a specific Lys residue. Please add viruses to the list of rhodopsin sources.

Response: We agree with your comment. Therefore, we have accordingly revised and cited a new reference as follows:

“Rhodopsins are light-driven seven-transmembrane proteins with a retinal chromophore that binds to a specific Lys residue and are found in prokaryotes, eukaryotes, and viruses^{4,5}” on page 3, lines 35-36.

⁵ Rozenberg, A. *et al.* Microbial Rhodopsins: The Last Two Decades. *Annual Review of Microbiology* **75**, 427–447 (2021).

Comment 5: Page 3, lines 44-45: “Unlike type-I and -II rhodopsins, the topology of *HeR* is invertedly embedded”
Please change to: “topology of *HeRs* is inverted, as compared to that of type-I and -II rhodopsins”

Response: We agree and have changed this according to your suggestion on page 3, lines 44-45.

Comment 6: Page 3, lines: Particularly, bacterial *HeRs*, similar to sensory rhodopsin (SR), lack ion pumping activity and have a long-lived photoactivated state”.

“Similar” here is confusing, and “particularly” has no sense. Please change to: “Bacterial *HeRs* lack ion pumping activity and have a long-lived photoactivated state, as do sensory rhodopsins (SRs) that belong to type I”.

Response: We agree and have changed this according to your suggestion on page 3, lines 46-47.

Comment 7: Page 3, line 52: “*HeR* functions remained largely unknown”
Please replace with: “they have not yet been tested experimentally”.

Response: We agree and have replaced this according to your suggestion on page 3, lines 52-53.

Comment 8: Page 3, line 58: “A closely adjacent gene”
A gene cannot be “closely adjacent”: it is either “adjacent” or “closely located”.

Response: We thank you for the comment. To make it easier for readers to understand, the sentences were modified as follows:

“In this study, *helR* and the gene adjacent to *helR* in the same operon were predicted to be continuously transcribed by a single promoter.” on page 3, lines 59-60.

Comment 9: Page 4, lines 76-77: “Crucial interaction positions between OcHeR and OmrDE were performed”
Positions cannot be performed; please rephrase to clarify what you mean.

Response: Thank you for pointing this out. The word “perform” made it strange to understand. Therefore, we have revised the text as follows:

“Interaction positions between OcHeR and OmrDE were investigated using protein–protein docking simulation and isothermal titration calorimetry (ITC) analysis. Afterward, we determined how the interaction affects OmrDE activity *in vitro* and *in vivo* in the absence and presence of light.” on page 4, lines 76-79.

Comment 10: Page 4, line 90-91: “suggesting that these patterns were identified only in prokaryotes”
Please change to “suggesting that these patterns occur only in prokaryotes”

Response: We agree and have changed this according to your suggestion on page 4, line 90.

Comment 11: Page 4, line 92: “Considering promoters, terminators, and closely neighboring genes...”
Do you mean “...and coding regions”?

Comment 12: Page 4, lines 92-93: “we classified the HeRs capable of transcribing *helR*”
This sentence makes no sense at all. The genes, including *helR* genes, are transcribed by RNA polymerases, not by heliorodopsin proteins.

Response: Comments (11 and 12) have led us to agree that these sentences may pose challenges for reader comprehension. Therefore, we have revised this information as follows:

“Frequently, the neighboring genes flanking *helR* were identified in the same operons. Thus, we classified HeRs into ten groups with each frequent neighboring gene:” on page 4, lines 90-92.

Comment 13: Page 4, line 98: “we investigate functional key residues that interacts to neighboring gene proteins”
Please change to: “we identified key residues responsible for interaction with proteins encoded by the neighboring genes”

Response: We agree and have changed this according to your suggestion on page 4, lines 97-98.

Comment 14: Page 4, lines 104-105: “Particularly, the number of amino acids in hydroxylic, aliphatic, aromatic, acidic, and basic amino acids were increased”
Please change to: “Particularly, the number of amino acids in hydroxylic, aliphatic, aromatic, acidic, and basic groups were increased”

Response: We agree and have changed this according to your suggestion on page 5, lines 105-106.

Comment 15: Page 5, line 108: “are more potential to have protein-protein interactions (PPIs)”
Please change to “are more likely to have protein-protein interactions (PPIs) than” and add than what.

Response: We agree with your comment and have accordingly revised the text to “are more likely to have protein–protein interactions (PPIs) than the non-co-transcription group” on page 5, lines 109-110.

Comment 16: Page 5, lines 109-110: “we aimed ABCT group because HeR may interact with ABCT as membrane protein similar to SR interacting with membrane proteins”
This is highly confusing. I understand that by the “ABCT group” the Authors mean such helR genes that have genes encoding ABCT as their neighbors, but the reader may think that the Authors mean ABCT genes themselves. Therefore, I suggest to use here and elsewhere something like “a group of ABCT-containing operons” instead of just “ABCT group”. Also, please change “aimed” to “focused on”.

Response: We agree with your comment and have accordingly revised “ABCT group” throughout the manuscript. Moreover, we have changed “aimed” to “focused on” on page 5, line 134.

Comment 17: Page 5, lines 112-113: “OcHeR exhibited protein sequence similarity with reported HeRs rather than with microbial rhodopsins”
This information is redundant: if OcHeR would be homologous to microbial rhodopsin, it would not be classified as heliorhodopsin! Please delete this sentence.

Response: We agree and have accordingly deleted this sentence.

Comment 18: Page 5, line 125: “functional critical motifs”
Please change to “functionally important motifs”

Response: We agree and have changed this according to your suggestion on page 6, line 152.

Comment 19: Page 5, lines 129-130: “the catalytic glutamate next to the Walker B motif in OmrE, the ATP-binding site in the NBD, remained canonical”
Please change to: “the canonical catalytic glutamate next to the Walker B motif in OmrE, the ATP-binding site in the NBD, is conserved”

Response: We agree and have changed this according to your suggestion on page 6, lines 156-157.

Comment 20: Page 5, lines 131-132: “phylogenetic tree among multidrug ABC exporters”
Please change to “phylogenetic tree of multidrug ABC exporters”

Response: We agree and have changed this according to your suggestion on page 6, lines 161.

Comment 21: Page 5, line 139: “pKa values of its counterion and retinal Schiff base were 2.883 and 11.385”
I doubt that it is reasonable to provide three decimal places for the constants; please reduce to one, i.e., 2.9 and 11.4. Also, please briefly explain why you attributed these constants to the counterion and retinal Schiff base, respectively.

Response: We thank you for your comment. We have briefly explained the counterion and the retinal Schiff base as follows:

“The wavelength of maximum absorbance (λ_{\max}) of OcHeR was 554 nm at neutral pH. From pH titration, the pKa value through the spectral red-shift of the λ_{\max} of OcHeR at acidic pH values was estimated to be 2.9, reflecting that the protonation of counterion (presumably E118) (Supplementary Figs. 9a,b). The pKa value through the deprotonation of the retinal Schiff base linked to K252 of OcHeR at alkaline pH values was estimated to be 11.4 based on the difference in absorbance of the deprotonated form of the retinal Schiff base (Supplementary Figs. 9c,d).” on page 6, lines 165-171.

Moreover, the pKa value of the retinal Schiff base of OcHeR was determined by difference absorbance (Supplementary Figs. 9c,d).

Comment 22: Page 5, lines 141-142: “The co-expression of OmrD and OmrE revealed high purity”
Please change to: “Western blotting revealed high purity of the corresponding proteins upon co-expression of OmrD and OmrE”

Response: We agree that your suggestion is better than the original sentence. Since Figure 3c for that sentence is the SDS-PAGE experiment, we have revised it according to your suggestion by changing western blot to SDS-PAGE on page 6, lines 172-173.

Comment 23: Page 6, line 147: “We investigated drug-screening”
Carried out drug screening?

Response: Thank you for pointing this out. We have changed “investigated” to “performed” on page 6, line 178.

Comment 24: Page 6, lines 150-152: “in the N43 strain harboring plasmids, that is, cell-plasmid [empty, OmrDE, and OmrD WT with OmrE D577Q as the canonical residue mutation associated with ATP binding in NBDs (OmrDE^m)]”

What is “cell-plasmid”? Also, it is unclear whether the N43 strain carried all three plasmids simultaneously, or these plasmids were tested in parallel. Please explain.

Response: Thank you for pointing this out. We agree with your comment and have revised this as follows:

“We carried out drug-screening for OmrDE in the *E. coli* N43 strain (*acrA* mutation) as a minimized cell line for affection drug susceptibilities²⁷ and further testing the minimum inhibitory concentration (MIC) of drugs (chloramphenicol, kanamycin, tetracycline, DAPI, and HO342) in the N43 strain harboring each plasmid [pET21b-empty, -OmrDE, and -OmrDE^m (OmrD WT and OmrE D577Q as the canonical residue mutation associated with ATP binding in NBDs)], termed cell-plasmid.” on page 6, lines 178-183.

Comment 25: Page 6, line 153: “were not significantly inhibited”
Please change to “were not strongly reduced”. Please note that the word “significantly” means the results of statistical hypothesis testing, which has not been done in this case.

Response: We agree and have changed the text according to your suggestion on page 6, line 184.

Comment 26: Page 6, line 159: “not only 3.85-fold more resistant”

Again, please reduce the number of decimal places to one, i.e., 3.9-fold. Please also do the same for other numeric estimates in the text.

Response: We agree with your comment and have accordingly revised the text throughout the manuscript.

Comment 27: Page 7, line 202: “accelerated compared with diffusion”.

Unclear, how charge interaction can be accelerated, and why should it be compared with diffusion? Please explain, and please also explain why you consider only long-distance charge interactions.

Response: We agree that the sentences are unclear to understand. Therefore, we have revised the text as follows:

PPI involves the proximity of two proteins via translational and rotational diffusions. An association rate constant enhanced by a long-range force (5–10 Å, electrostatic interaction) is stronger than that enhanced by short-range forces (hydrophobic and Van der Waals), accelerating diffusion due to electrostatic attraction between proteins²⁶. Charges can affect an electrostatic interaction and be specific to each protein²⁷. The “positive-inside” rule for membrane proteins is that positively charged residues are more frequently found on the cytoplasmic side of membrane proteins than on the extracellular side^{10,28}. The charged residues of HeRs are important for binding to enzymes^{15,16}. Considering the characteristic of electrostatic interaction, we suggest that charged residues in OcHeR may affect PPIs.” on page 5, lines 111-119.

Comment 28: Page 7, line 217: “HeRs showed a greater conservative positively charged residues”

Please change to “HeRs showed a greater conservation of positively charged residues”, and also change “implying” to “suggesting” in the next line.

Response: We agree and have changed the text according to your suggestion on page 5, lines 129-131.

Comment 29: Page 9, line 290: “were similar to and lower than those of OmrDE alone”

Please change to ““were similar to, and lower, respectively, than those of OmrDE alone”

Response: We agree and have changed the text according to your suggestion on page 9, line 274.

Comment 30: Page 10, line 322: “The photocycle is a photochemical reaction of conformational changes and molecular events”

Please change to “The photocycle is a series of photochemical reactions of the chromophore and conformational changes in the apoprotein”

Response: We agree and have changed the text according to your suggestion on page 9, lines 306-307.

Comment 31: Page 10, line 345: “encoded in the same direction as the single promoter”

Please change to “encoded in the same direction under a single promoter”

Response: We agree and have changed the text according to your suggestion on page 10, line 329.

Comment 32: Page 12, lines 398-399: *“potential applications to the target are indirect because of its functions as a channel, pump, and enzyme-fused protein”*

Unclear, what the Authors mean here. Each microbial rhodopsin used in optogenetics has its unique mode of action, and those that exhibit, e.g., channel activity, do not act as photoregulated enzymes.

Response: Thank you for pointing this out. We agree that the sentence is unclear; thus, we have removed the sentence because it does not seem necessary to mention.

Reviewer #2 (Remarks to the Author):

Heliorhodopsin (HeR) is a novel member of rhodopsin family. The discovery of this member, which is widely distributed in microorganisms and lower eukaryotes, came as a great surprise to many researchers. Despite its widespread distribution, the physiological role of HeR has not been fully clarified. In this study, Cho et al. reported that some HeRs make complex with the multidrug efflux transporters and can elevate the drug transport activities under illumination. This finding is very interesting and deserves to be widely known. However, for many readers, it is not so easy to understand the content of current manuscript. Thus, I'd like to request the authors to consider the following points:

Response: We appreciate your acknowledgment of the novelty and significance of our findings. Your positive assessment greatly encourages us, and we are pleased to learn that you find our study to be of interest and deserving of wider recognition. In response to your suggestions to enhance the manuscript's accessibility, we have carefully considered your comments and were committed to improving the clarity of our content. We have revised the manuscript to ensure that it is more comprehensible to a broader audience. Your comments undoubtedly contributed to the overall improvement of the paper.

Comments from Reviewer #2

Comment 1: *Structure of the manuscript*

The most important data should be 1) increased drug transport activity of a multidrug efflux transporter OmrDE by the complex formation with a heliorhodopsin named OcHeR and 2) further increase of the OmrDE activity under illumination. However, these data are briefly described after a lengthy description about other data, such as a) the amino acid sequence comparison among many HeRs, b) predicted structure of the multidrug transporter, and c) the ITC data to explore complex formation and its contact points, and so on. To understand these "other data", the readers need to refer so many supporting figures and experimental procedures. I think this manuscript could be revised to something simpler and more readable.

Response: Thank you for pointing this out. We agree with this comment. We have revised the overall manuscript and main figures so that the important data can be read as quickly as possible.

Comment 2-1: *Data showing the activity of OmrDE*

Since these data are the hearts of this study, the authors should explain their details.

For Fig. 2d, the authors should explain the following three points:

(a) why the fluorescence intensity decreases when DAPI is transported into the inverted membrane vesicles (IMVs), (b) why the fluorescence decrease also occurs in the "Empty" IMV, whose membrane does not contain OmrDE, and (c) why "Empty" IMV has lower fluorescence intensity compared to "OcHeR" IMV.

Response: We appreciate your detailed comment. We agree that three reasons should be explained. Therefore, we have revised the text as follows:

"To evaluate drug transportation capacity, we performed a DAPI translocation assay in an inverted membrane vesicle (IMV). The fluorescence intensity for time-tracing by adding ATP can be attributed to two reasons: 1) OmrDE transports DAPI into the IMV from the reaction solution. 2) The fluorescence intensity of DAPI transported inside is relatively reduced owing to interference with the membrane (Supplementary Figs. 11a,b). OmrDE showed a higher transportation capacity compared to the empty vector, OmrDE^m, and OcHeR (Fig. 3f). As endogenous multi-drug ABCTs of *E. coli* also transport DAPI, the fluorescence intensity of the empty vector may be decreased. Interestingly, the capacity of OcHeR was lower than that of the empty vector, suggesting that the embedding positions of the endogenous multi-drug ABCTs in the lipid bilayer are limited by OcHeR overexpression." on page 7, lines 193-202.

Comment 2-2: *For Fig. 2e and Fig. 6a, the authors should show the raw data used to calculate the V₀ values at each ATP concentration.*

Response: We have added this in Supplementary Data and mentioned this in Table 1 legend as follows:

"V₀ values at each ATP concentration are provided in Supplementary Data."

Comment 2-3: For Fig. 6e, I could not understand why distinct difference between "DE+WT Dark" and "DE+WT Light" was appeared in the cell viability. These two traces should have been calculated from the data in Fig. S7j, which indicates little difference in the cell viability under Dark and Light conditions.

Response: I apologize for any confusion caused by the difference between Fig. 5e (revised Fig. 6e). and Fig. S10j (revised Fig. S7j).

To clarify, both figures indeed utilize the same dataset; the difference lies in the representation of the x-axis. In Fig. S10j, a logarithmic transformation (\log_{10}) of the DAPI concentration was applied to the x-axis. This transformation was employed to enhance the visibility of the curve shape and improve the fit quality when concentrations cover a wide range. Consequently, the IC_{50} values were calculated based on this log-transformed x-axis while the y-axis reflects cell viability.

I acknowledge that Fig. S10j requires manual recalculation by the reader to convert the \log_{10} (DAPI) concentrations to DAPI concentrations. Therefore, we represented the number (not the logarithmic number) in Fig. 6e. To address this potential source of confusion, we have now explicitly stated in the legend of Fig. 5e the following information:

"Non-linear fits were obtained from Supplementary Figs. 10h–m where the x-axis modified logarithmic number was changed to the non-logarithmic number."

This additional information provides clarity on the rationale behind the distinct appearance of the curves in Fig. 5e and align the understanding with the underlying data in Fig. S10j. We appreciate the reviewer's meticulous comment of the data presentation, and we trust that this clarification addresses any potential ambiguities.

Comment 3: Comparison of amino acid sequences of heliorhodopsins

Significance of Fig. 2c, Fig. 4c, and other similar figures are not clear. Moreover, it is probably difficult for most readers to understand how these figures were prepared. Based on these data, the authors determined the amino acids to be mutated for ITC experiments and activity measurements of OmrDE. Instead of this method, I think that the authors should simply use the results of docking simulation of OmrDE and OcHeR (Fig. 3f, Fig. 5a) to determine the residues to be mutated.

Response: Sorry for the confusion regarding this data. We totally agree with your comment. Therefore, the analysis process was illustrated with figures to make it easier for readers to understand, and detailed information is provided in the legend of Supplementary Fig. 1. Moreover, we have revised the text as follows:

"Moreover, we identified the residues responsible for interaction with proteins encoded by the neighboring genes. We analyzed the differences of amino acid frequency per HeR at each position in OcHeR by studying the amino acids of each HeR in a co-transcription group (*heIR* and the neighboring gene in operon) and the classified ten groups to a non-co-transcription group (*heIR* alone in operon) (detailed analyzing procedures are in Supplementary Fig. 1)." on page 4, lines 97-101.

We mainly chose the residues for mutations based on the docking simulation and references on the effect of mutating the charged residues. We were unable to choose residues for mutation solely through the docking simulation owing to an accuracy.

The accuracy of the protein structures is essential for the success of modeling protein complexes. The structures of OcHeR and OmrDE were modeled by HeR-48C12 and TmrAB as templates using Swiss-Model, respectively. Thus, we did not consider the docking simulation results to be 100% reliable since the structures were predicted. Furthermore, there would be various interaction sites, but it seemed difficult to mutate residues and experiment with numerous mutants.

For these reasons, we considered three possibilities: 1) Polar interaction at the interface between OcHeR and OmrDE via the docking simulation; 2) charged and aromatic residues at the interface involved in polar and hydrophobic interactions, respectively, may be binding sites; 3) the binding sites between HeR and enzyme were residues on the cytoplasmic side in HeR according to reported HeRs. The important binding sites may be on the cytoplasmic side.

Comment 4: ITC experiments

Because of the weak binding, saturation of binding was not observed in the most data (Fig. 3b, Fig. S10). Thus, it is questionable how correct the values of thermodynamic parameters determined from these data are. As noted above, the most important result in this paper is that binding of OcHeR alters the activity of OmrDE (Fig. 6a-f). Therefore, the ITC results are not essential. I'd like the authors to reconsider whether they can have a credible discussion based on the ITC data.

Response: We appreciate the insightful comments.

The challenge posed by weak binding in protein–protein interactions, where the dissociation constant (K_d) is typically in the micromolar range is acknowledged. In our study, the K_d value of approximately 100 μ M reflects this characteristic of OcHeR–OmrDE interactions. We predict that achieving a sigmoidal curve between OcHeR and OmrDE using ITC analysis would require the injection of a high concentration of OcHeR into OmrDE, potentially approximately 4 mM (~120 mg/mL). However, preparing such high concentrations of proteins is experimentally challenging and has inherent limitations.

To address the concerns raised about the saturation curve, we emphasize that we performed multiple repetitions of the ITC experiments ($n = 2 \sim 5$) under consistent or slightly modified conditions. The resulting thermodynamic parameters exhibited a high degree of consistency, supporting the reliability of our data. Recognizing the limitations inherent in achieving perfect saturation in OcHeR–OmrDE interactions, we have relocated the discussion on thermodynamic parameters to the discussion section, away from the results section.

In our study, the comparison of K_d values between OcHeR WT and its mutants was a critical aspect. Despite maintaining a consistent ITC analysis methodology, even a single residue replacement in OcHeR resulted in divergent outcomes between the WT and mutants. Notably, mutants that did not bind to OmrDE exhibited distinct results. Therefore, the K_d values obtained for OcHeR and OmrDE in this study are deemed comparable and relevant.

Our decision to perform ITC analysis before *in vitro* and *in vivo* experiments aimed to validate the binding between OcHeR and OmrDE in their native state without any modifications or labeling. Its label-free nature and non-destructive properties allow for the study of biomolecular interactions in their native state, avoiding potential artifacts introduced by modifications or labeling. This approach aligns with the advantages of ITC analysis, while acknowledging the inherent limitations in achieving perfect saturation in protein–protein interactions.

We hope this clarification addresses the concerns raised and underscores the rationale behind our experimental choices. In addition, we have mentioned the limitations as follows:

“Owing to the limitations posed by weak binding in OcHeR–OmrDE interactions, perfect saturation was not achieved; however, the K_d values obtained for OcHeR and OmrDE in this study are comparable and relevant because the experimental methods for WT and mutants were identical.” on page 8, lines 263–266.

Comment 5: Swiss-model structure of OmrDE

To determine the type of ABC transporter to which OmrDE belongs, the authors compared the Swiss-model structure of OmrDE with the crystal structures of ABC transporters already reported (Fig. S4). The Swiss-model structure largely depends on the template structure. Probably, the authors chose the TmrAB structure as the template based on its high amino acid sequence similarity to OmrDE (Fig. S5). Thus, without comparing the structures, the authors were able to determine the type of OmrDE, which should belong to the same group as TmrAB.

Response: We considered ABC transporter as the template based on the identity, GMQE, and QSQE scores. 1) Identity score indicates the percentage of identical amino acid residues between the target sequence and the template structure. 2) The Global Model Quality Estimate (GMQE) score is a quality estimate that combines properties from the target–template alignment and the template structure. 3) The Quaternary Structure Quality Estimate (QSQE) scores indicate the expected accuracy of the interchain contacts required for a model built based appropriate alignment and template.

In addition, we also considered the various conformations of ABC transporter: Inward-facing narrow (IF_{narrow}), Inward-facing wide (IF_{wide}), occluded, and outward-facing (OF) conformations. We assumed that the conformation of purified OmrDE is the narrow separation of NBDs under apo condition (IF_{narrow}) rather than the large separation of NBDs (IF_{wide}). We have mentioned the detailed explanation in the Supplementary Discussion as follows:

“To investigate protein–protein interactions between purified OmrDE and OcHeR using ITC analysis, we considered

various conformations of ABCTs. ABCTs undergo conformational changes, transitioning from inward-facing (IF) to outward-facing (OF) conformations through ATP binding and hydrolysis¹. The IF conformation comprises two distinct forms: the narrow separation of NBDs under apo condition (IF_{narrow}) and the wide separation of NBDs (IF_{wide}). Notably, substrate binding is specifically associated with the IF_{wide} conformation². It is conceivable that the IF_{narrow} conformation in purified protein may be more frequent than the IF_{wide} conformation in the absence of substrate due to release and dissociation. Consequently, when the purified protein is in a substrate-deficient solution, it may predominantly adopt the IF_{narrow} conformation. Therefore, we assumed that the conformation of purified OmrDE is the IF_{narrow} conformation rather than the IF_{wide} conformation.”

¹ Locher, K. P. Mechanistic diversity in ATP-binding cassette (ABC) transporters. *Nat Struct Mol Biol* **23**, 487–493 (2016).

² Hofmann, S. *et al.* Conformation space of a heterodimeric ABC exporter under turnover conditions. *Nature* **571**, 580–583 (2019).

Overall, we chose the TmrAB IF_{narrow} conformation structure based on three scores (identity, GMQE, and GSQE) and conformations of the ABC transporter. Nevertheless, we did not show a figure comparing the structures between OmrDE and TmrAB; therefore, we have added structural comparison figures (Figs. 3a,b and Supplementary Fig. 7c). Moreover, we have mentioned the identity, GMQE, and GSQE scores in the Supplementary Discussion.

Reviewer #1 (Remarks to the Author):

The Authors have considerably rewritten their manuscript according to my comments and those by the fellow Reviewer. I recommend the revised version for publication, but have the following suggestions for the title, abstract, and introduction, which are aimed to further improve its clarity:

Page 1, line 1 (the title): "Novel insights into particular..."

I think these words are unnecessary and should be deleted. The words "novel insights" are just noise carrying no information – if this manuscript had no novel insights, the Authors would not submit it. To the best of my knowledge, this is the first report on photoregulation of ABC transporter mediated by rhodopsin, so there is no need to use "particular", another word that lacks information. Ditto for the word "specific". On the other hand, the Authors would want to emphasize that this rhodopsin was a heliorhodopsin, as heliorhodopsins are a recently discovered protein family still poorly characterized. So, my suggestion for the title is: "Heliorhodopsin-mediated light-modulation of ABC transporter."

Page 1, lines 15-18: "Herein, we expand on the potential functions of Particular helioRhodopsin mediated Specific light-Modulation (PRISM) in relationship with Omithinimicrobium cerasi HeR (OcHeR) and heterodimeric multidrug resistance ATP-binding cassette transporters (OmrDE)."

This is way too convoluted and difficult to understand. Why not say simply: "Herein, we investigate photomodulation of heterodimeric multidrug resistance ATP-binding cassette transporters (OmrDE) mediated by Omithinimicrobium cerasi HeR (OcHeR)?"

Page 3, lines 27-34: "Since Earth's formation... <...> for light absorption4".

I think the entire 1st paragraph of Introduction should be deleted as being irrelevant for the topic of this study.

Page 3, lines 43-44: "Heliorhodopsin (HeR) is a novel rhodopsin discovered 5 years ago, forming a distinct clade separate from type- rhodopsin".

Dear Authors, please check your usage of the singular and plural noun forms throughout the manuscript! A single heliorhodopsin cannot form an entire clade, nor can a single type-I rhodopsin!

Reviewer #2 (Remarks to the Author):

I appreciate the authors for addressing to my comments. In the revised version, I found several points that I'd like to request the authors to consider.

1) ITC data of binding between OcHeR and OmrDtcEtc (Fig. S12c):

In the corresponding data between OcHeR and OmrDE, endothermic reaction was observed (Fig. S12b). However, the opposite exothermic reaction was observed between OcHeR and OmrDtcEtc. This difference is very drastic. Thus, the authors should discuss this difference.

2) Manuscript, page 5, line 120: "negative" should be "positive".

3) Manuscript, page 6, line 183: The description of "The MIC values of the antibiotics for cells were not strongly reduced." is not clear and might be better modified as follows: "The MIC values of the antibiotics (Cm, Km, Tet) for the cells were not strongly elevated by OmrDE and OmrDEm.".

4) SI, page 24, caption of Fig. S12: "W171A" should be "W170A".

Author's point by point response to the reviewers' comments

Manuscript ID: NCOMMS-23-44201B

Title: Heliorhodopsin-mediated light-modulation of ABC transporter

Our point-by-point response to reviewers' comment are detailed as follows.

Responses are presented subsequent to the **Comments**. And the revised texts on our manuscript are **highlighted**.

Reviewer #1 (Remarks to the Author):

The Authors have considerably rewritten their manuscript according to my comments and those by the fellow Reviewer. I recommend the revised version for publication, but have the following suggestions for the title, abstract, and introduction, which are aimed to further improve its clarity:

Response: Thank you sincerely for your thorough review and constructive feedback on our revised manuscript. We greatly appreciate the time and effort you have dedicated to providing insightful comments. Moreover, we are delighted to hear of your recommendation for publication. We welcome your suggestions for improving the title, abstract, and introduction further. Your continued support and guidance are invaluable to us, and we are grateful for the opportunity to incorporate your expertise into our manuscript.

Comments from Reviewer #1

Comment 1: Page 1, line 1 (the title): "Novel insights into particular..."

I think these words are unnecessary and should be deleted. The words "novel insights" are just noise carrying no information – if this manuscript had no novel insights, the Authors would not submit it. To the best of my knowledge, this is the first report on photoregulation of ABC transporter mediated by rhodopsin, so there is no need to use "particular", another word that lacks information. Ditto for the word "specific". On the other hand, the Authors would want to emphasize that this rhodopsin was a heliorhodopsin, as heliorhodopsins are a recently discovered protein family still poorly characterized. So, my suggestion for the title is: "Heliorhodopsin-mediated light-modulation of ABC transporter."

Response: Thank you for your suggestion. We agree with your comment and have accordingly revised the title.

Comment 2: Page 1, lines 15-18: "Herein, we expand on the potential functions of Particular helioRhodopsin medlated Specific light-Modulation (PRISM) in relationship with Omithinimicrobium cerasi HeR (OcHeR) and heterodimeric multidrug resistance ATP-binding cassette transporters (OmrDE)."

This is way too convoluted and difficult to understand. Why not say simply: "Herein, we investigate photomodulation of heterodimeric multidrug resistance ATP-binding cassette transporters (OmrDE) mediated by Omithinimicrobium cerasi HeR (OcHeR)"?

Response: We agree with your comment. Thus, we have accordingly revised the sentence and removed "PRISM" throughout the manuscript.

Comment 3: Page 3, lines 27-34: "Since Earth's formation... <...> for light absorption4".

I think the entire 1st paragraph of Introduction should be deleted as being irrelevant for the topic of this study.

Response: We agree and have deleted the 1st paragraph of introduction.

Comment 4: Page 3, lines 43-44: “Heliorhodopsin (HeR) is a novel rhodopsin discovered 5 years ago, forming a distinct clade separate from type- rhodopsin”.

Dear Authors, please check your usage of the singular and plural noun forms throughout the manuscript! A single heliorhodopsin cannot form an entire clade, nor can a single type-I rhodopsin!

Response: Thank you for pointing this out. We have checked and revised the usage throughout the manuscript.

Reviewer #2 (Remarks to the Author):

I appreciate the authors for addressing to my comments. In the revised version, I found several points that I'd like to request the authors to consider.

Response: Thank you for your review and for taking the time to provide your feedback on our manuscript. We sincerely appreciate your acknowledgment of the efforts we've made to address your comments. Your input has been invaluable in improving the quality of our work. We welcome your suggestions for further consideration and will carefully review them to ensure the continued enhancement of our manuscript.

Comments from Reviewer #2

Comment 1: 1) ITC data of binding between OcHeR and OmrD^{tcE^{tc}} (Fig. S12c):

In the corresponding data between OcHeR and OmrDE, endothermic reaction was observed (Fig. S12b). However, the opposite exothermic reaction was observed between OcHeR and OmrD^{tcE^{tc}}. This difference is very drastic. Thus, the authors should discuss this difference.

Response: We appreciate the insightful comments. We have discussed this difference as follows:

"Interestingly, the ΔH value of OcHeR for OmrD^{tcE^{tc}} indicated an exothermic reaction, whereas that of OcHeR for OmrDE indicated an endothermic reaction (Table 2). ABCTs undergo conformational changes through ATP binding and hydrolysis in NBDs¹⁵, suggesting that OmrD^{tcE^{tc}} forms a different heterodimeric structure from OmrDE. Due to this difference, enthalpic factor contributed from unfavorable ($\Delta H > 0$) to favorable ($\Delta H < 0$) non-covalent interactions, thereby forming new bonds and facilitating the release of energy, as well as influencing the entropy change." on page 7, lines 221-227.

Comment 2: 2) Manuscript, page 5, line 120: "negative" should be "positive".

Response: Thank you for pointing this out. We have changed "negative" to "positive".

Comment 3: 3) Manuscript, page 6, line 183: The description of "The MIC values of the antibiotics for cells were not strongly reduced." is not clear and might be better modified as follows: "The MIC values of the antibiotics (Cm, Km, Tet) for the cells were not strongly elevated by OmrDE and OmrDE^m".

Response: We agree that your suggestion is better than the original sentence. We have revised it according to your suggestion on page 6, lines 176-178..

Comment 4: 4) SI, page 24, caption of Fig. S12: "W171A" should be "W170A".

Response: Thank you for pointing this out. We have revised this.